# The Absorption Ångström Exponent of black carbon: from numerical aspects

Chao Liu[1,2,*], Chul Eddy Chung[3], Yan Yin[1,2], and Martin Schnaiter[4]

[1]Collaborative Innovation Center on Forecast and Evaluation of Meteorological Disasters, Nanjing University of Information Science & Technology, Nanjing 210044, China
[2]Key Laboratory for Aerosol-Cloud-Precipitation of China Meteorological Administration, School of Atmospheric Physics, Nanjing University of Information Science & Technology, Nanjing 210044, China
[3]Desert Research Institute, Reno, 89512, USA
[4]Karlsruhe Institute of Technology, Institute of Meteorology and Climate Research, 76021, Karlsruhe, Germany

*Correspondence to*: Chao Liu (chao_liu@nuist.edu.cn)

**Abstract.** The Absorption Ångström Exponent (AAE) is an important aerosol optical parameter used for aerosol characterization and apportionment studies. The AAE of black carbon (BC) particles is widely accepted to be 1.0, although observational estimates give a quite wide range of 0.6~1.3. With considerable uncertainties related to observations, a numerical study is a powerful method, if not the only one, to provide a better and more accurate understanding on BC AAE. This study calculates BC AAE using realistic particle geometries based on fractal aggregate and an accurate numerical optical model (namely the Multiple-Sphere T-Matrix method), and considers only bulk properties of an ensemble of BC particles following lognormal size distributions. At odds with the expectations, BC AAE is not 1.0, even when BC is assumed to have small sizes and a wavelength independent refractive index. With a wavelength independent refractive index, the AAE of fresh BC is approximately 1.05, and is relatively insensitive to particle size. For BC with geometric mean diameters larger than 0.12 µm, BC AAE becomes smaller when BC particles are aged (compact structures or coated by other non-absorptive materials). For coated BC, we prescribe the coating fraction variation based on a laboratory study, where smaller BC cores are shown to develop larger coating fractions than bigger BC cores. For both Compact and Coated BC, the AAE is highly sensitive to particle size distribution, ranging from approximately 0.8 to even over 1.4 with wavelength-independent refractive index. When the refractive index is allowed to vary with wavelength, a feature with observational backing, the BC AAE may show an even wider range. For different BC morphologies, we derive simple empirical equations on BC AAE based on our numerical results, which can serve as a guide for the response of BC AAE to BC size and refractive index. Due to its complex influences, the effects of BC geometry is better to be discussed at certain BC properties, i.e. known size and refractive index.

## 1 Introduction

The Absorption Ångström Exponent (AAE) is an aerosol optical property, and describes the wavelength variation of aerosol absorption. Because aerosol absorption normally decreases exponentially with wavelength over the visible and near-infrared spectral region (Ångström, 1929; Bond, 2001; Lewis et al., 2008), the AAE is defined as:

$$C_{abs}(\lambda) = C_o \lambda^{-AAE} \qquad \text{or} \qquad ln(C_{abs}(\lambda)) = ln(C_o) - AAE \; ln(\lambda) \qquad (1)$$

where $\lambda$, $C_{abs}$ and $C_o$ denote wavelength, the aerosol absorption coefficient, and a wavelength-independent constant (that equals the absorption coefficient at the wavelength of 1µm). Instead of the absorption coefficient, some studies use the aerosol absorption optical depth in Eq. (1), because the two are proportional. The AAE describes absorption variation with respect to wavelength, and is significantly influenced by particle size, shape, and chemical composition (Scarnato et al., 2013;

Schuster et al., 2015; Li et al., 2016). The chemical composition of aerosol material determines the wavelength-dependent refractive index.

AAE has been widely used for aerosol characterization studies (Russell et al., 2010; Giles et al., 2012). The basis for the AAE in aerosol characterization is that AAE is assumed to be a specific property of each aerosol species. For example, black

carbon (BC) aerosols have AAEs around 1.0, and organic aerosols and dust have higher AAE values (Kirchstetter et al., 2004; Russell et al., 2010). Thus, the AAE of an aerosol sample close to 1.0 is considered as BC rich aerosols from fossil fuel burning, and larger AAE values are understood to indicate aerosols from biomass/biofuel burning or dust (Russell et al., 2010). AAE has also been quantitatively used to separate brown carbon (BrC) absorption from BC absorption (Kirchstetter and Thatcher, 2012; Lu et al., 2015). In the studies of BC/BrC absorption separation, the BrC absorption from biomass

burning aerosols is usually retrieved by assuming that BrC contributes no absorption at near-infrared wavelengths and that BC has an AAE of 1.0 (Ganguly et al., 2005; Kirchstetter and Thatcher, 2012; Lu et al., 2015). Lack and Langridge (2013) quantified the effect of a specified BC AAE value on the absorption apportionment between BC and BrC, and demonstrated the importance of BC AAE. Additional uses of AAE include aerosol color. A recent numerical study done by Liu et al. (2016a), for example, confirms that AAE largely controls the color of aerosols in the ambient atmosphere. They found that

aerosols have a visual of brown color if its AAE is larger than approximately 2. In another use, aerosol spectral light absorption is an important parameter for the assessment of the radiation budget of the atmosphere (Schmid et al., 2006).

BC, also known as soot, is an important aerosol species emitted from incomplete combustion of fossil fuel, biofuel and biomass (Bond and Sun, 2005; Bond et al., 2013; Chakrabarty et al., 2014), and exhibits significant variations on its physical

and chemical properties due to differences in fuels and combustion conditions (Schnaiter et al., 2006; Bahadur et al., 2012; Reddington et al., 2013). Does BC indeed have an AAE of 1.0? The AAE of BC from combustion is widely accepted and used as 1.0 when the particles exist alone and have not experienced atmospheric aging processes (Bergstrom et al., 2002; 2003; Schnaiter et al., 2003; Lawless et al., 2004; Bond and Bergstrom, 2006). If BC particles are much smaller than the

incident light wavelength and have a wavelength-independent refractive index, the Rayleigh approximation does theoretically derive an AAE of 1.0 (Moosmüller and Arnott, 2009). Atmospheric BC particles are generally small compared to the wavelengths of visible light, but it is uncertain that all the BC particles clearly fall into the Rayleigh regime, let alone the uncertainty on the wavelength dependence of the BC refractive index. Actually, real BC aggregates in the ambient atmosphere may just fall at the edge of the Rayleigh region at visible and near infrared wavelengths. Figure 1 visualizes the extent to which the Rayleigh approximation holds for spheres by comparing with the exact Mie results. In Fig. 1, the x-axis represents the particle size parameter, defined as $2\pi r/\lambda$ with r and $\lambda$ being radius and wavelength, respectively. We can see from Fig. 1 that the Rayleigh approximation is only valid for absorption if the size parameter is less than 0.2. To be more specific, an aerosol sphere with a radius of 20 nm corresponds to a size parameter of approximately 0.13 at a wavelength of 1000 nm, at which the Rayleigh and Mie results agree quite well. However, if the wavelength decreases to 300 nm, the size parameter of the same sphere increases to 0.42, and the Rayleigh approximation underestimates the absorption by over 10%. BC particles from combustion processes are aggregates composed of hundreds or even thousands of spherical monomers with radii in the range of about 5 to 25 nm, and even these monomers lies out of the Rayleigh regime. The BC aggregate can easily has radii over 100 nm, which makes the Rayleigh theory even less applicable to BC absorption calculation. Not that BC size will be discussed in details in the next section.

When the true BC AAE is in doubt, one can alternatively investigate its AAE by measuring the absorption of BC particles in the atmosphere, which turns out to be more challenging. BC AAE has been experimentally investigated in numerous studies (e.g., Schnaiter et al., 2003; Kirchstetter et al., 2004; Bahadur et al., 2012; Chung et al., 2012). In a laboratory study, Schnaiter et al. (2003) found diesel soot to have an AAE of 1.1 and spark-generated carbon nanoparticles to have an AAE of 2.1. The different AAE values were mainly attributed to differences in the wavelength dependence of the refractive index for the two BC materials. In the atmosphere, however, BC particles always co-exist with other aerosol particles. Non-BC particles can affect the total aerosol AAE by containing BrC or mineral dust (which have higher AAE values) and also by coating BC. Coating of BC amplifies the BC absorption, and the amplification of BC absorption is dependent on wavelength. Kirchstetter et al. (2004) measured the absorption of particles near a roadway or inside a tunnel, and, after extracting organic carbon (including absorptive BrC), found the AAE to be 0.6~1.3. The locations Kirchstetter et al. (2004) chosen would have yielded AAE without much interference with BrC or coating. However, Kirchstetter et al. (2004) used filter-based instruments to measure the absorption. Filter-based absorption instruments are susceptible to multiple artefacts such as optical interactions between the concentrated particle themselves and that of the particles with the filter substrate (Moosmüller et al., 2009). In addition, filter deposition may alter particle shapes and size distributions greatly (Subramanian et al., 2007), which affects aerosol absorption properties significantly (Li et al., 2016). Weingartner et al. (2003) and Arnott et al. (2005), for example, attempted to address these artefacts. The available correction schemes are not available for every type of BC and furthermore not optimized for adjusting AAE. Chow et al. (2009) showed that Particle Soot Absorption Photometer (PSAP), after an absorption artefact correction, gave AAE values about 20% less than a filter-free photoacoustic

instrument for the aerosols at Fresno, California. Gyawali et al. (2012) and Chakrabarty et al. (2013) generated BC dominated particles by burning oil, and measured the absorption using filter-free photoacoustic instruments. The estimated AAE ranged from 0.8 (kerosene soot) to 0.95~1.1 (mustard oil soot). Because there must be some absorptive organic aerosols (e.g., BrC) in these aerosol samples that have a much larger AAE, the indication from these two studies is that BC

AAE is lower than 0.8~1.1.

In addition, AAE values could differ due to differences in AAE calculations. For example, if the absorptions at two wavelengths are observed, the AAE can be approximated by:

$$AAE \quad = -\frac{\ln(C_{abs1}/C_{abs2})}{\ln(\lambda_1/\lambda_2)} \qquad\qquad\qquad (2)$$

where $C_{abs1}$ and $C_{abs2}$ are the absorption coefficients at the wavelengths of $\lambda_1$ and $\lambda_2$, respectively. Due to the variation on different ranges of wavelengths, the AAE approximated by Eq. (2) becomes quite sensitive to the choice of observational wavelengths (Moosmüller and Chakrabarty, 2011).

Whether BC AAE is exactly 1.0 or not is an issue we will address in this paper. Another issue is whether BC coated with

non-absorptive material would have the same AAE as uncoated BC. In the aforementioned BC/BrC absorption separation studies (Kirchstetter and Thatcher, 2012; Lu et al., 2015), the AAE for uncoated BC was implicitly assumed to be the same as that for coated BC. Lack and Cappa (2010) used a core-shell Mie code to investigate how BC changes its AAE value with respect to coating. At realistic particle sizes, they showed that the BC AAE increases to 1.4~1.6 after coating. They computed BC AAE using a group of BC particles where the cores were specified to have a lognormal size distribution. In

their study, coating volume fraction was assumed to be fixed to all the particles, an assumption that has no experimental, observational or theoretical support. When BC particles grow in size by coating, the particle growth is governed by condensation. Theoretically, condensation reduces the diameter spread between big particles and small particles over time (Seinfeld and Pandis, 2016), on the contrary to the assumption by Lack and Cappa (2010). Schnaiter et al. (2005) coated BC particles with secondary organic aerosol material in a lab, and found that the coating increases particle sizes, while it reduces

geometric standard deviation (see Fig. 5 of their paper), as predicted by theoretical calculation of condensation process. Their work provided a meaningful experimental dataset to derive coating fraction variation of BC particles. Furthermore, Schnaiter et al. (2005) also measured the absorption of coated BC particles at 450, 550 and 700 nm (see Fig. 9 of their paper), and from these three wavelengths we see that coating does actually decrease BC AAE (from approximately 1.1 to 0.8), thereby contradicting the coated BC AAE estimates in Lack and Cappa's (2010) study. This indirectly indicates that a fixed

coating fraction for different-sized BC may be problematic.

Although observations do not give a clear value of BC AAE, it is safe to say that even accurate observations do not strongly support the theoretical constant of 1.0 for BC AAE. The fact alone that there are different types of soot particles (associated

with different refractive indices) points to a range of BC AAE instead of a fixed value. Furthermore, it is not clear if the real BC AAE is 1.0 on the average. Despite all these uncertainties, BC AAE has been assumed to be 1.0 in many studies (Lack and Langridge, 2013; Moosmüller et al., 2009; Lack et al., 2008; Kirchstetter et al., 2004; Lewis et al., 2008). Meanwhile, numerical studies on BC AAE did not show systematic nor conclusive results to improve our understanding on them (Li et
al., 2016; Lack and Langridge, 2013).

This study presents a systematic numerical investigation on the AAE of BC particles, and decomposes the AAE influence into that due to each particle microphysical property (e.g. shape, size, refractive indices, and internal mixing). The paper is organized as follows. The properties of BC particles used for absorption simulations are discussed in Section 2, and Section
3 presents the AAE simulations and decomposes the AAE influences. Section 4 concludes this study.

## 2 BC properties

The absorption of a single particle or an ensemble of particles can be accurately calculated if the particle shape, size, and refractive index are known. With a large amount of observations on BC microphysical and optical properties made in the past decades (Sorensen, 2001; Bond and Bergstrom, 2006), there is much less uncertainty in estimating shape, size, and
15 refractive index for BC compared with AAE estimation. The present study embarks on numerical calculations of the optical properties of an ensemble of BC particles. There have been numerous calculations of BC optical properties on a single wavelength (Sorensen, 2001; Liu and Mishchenko, 2005; 2007; Smith and Grainger, 2014), and the novelty of the present study is considering much wider but realistic ranges of BC properties (especially the particle shape and coating fraction) and focusing on its AAE systematically. The following subsections discuss BC geometry, size, and refractive index, and explain
how these properties are treated in the simulations herein.

### 2.1 BC geometry

BC particles exist in the form of aggregates with hundreds or even thousands of small spherical particles, called monomers. The concept of the fractal aggregate (FA) shows great success and wide applications on representing realistic BC geometries (Sorensen, 2001). The FA is mathematically described by the statistic scaling rule in the form of:

$$N = k_f \left(\frac{R_g}{a}\right)^{D_f} \tag{3}$$

where $N$ is the number of spherical monomers in an aggregate, $a$ the average radius of the monomers, $R_g$ the radius of gyration, $k_f$ the fractal prefactor, and $D_f$ the fractal dimension. Based on Eq. (3), as $k_f$ or $D_f$ increases, a relatively small $R_g$ is resulting, which corresponds to a relatively compact particle. The non-idealized factors, such as overlapping or necking among monomers, nonsphericity, and monomer size dispersion, do exist in reality, and their effects on BC optical properties
have been extensively studied (Skorupski and Mroczka, 2014; Yon et al., 2015; Wu et al., 2016; Liu et al, 2015a; Liu et al.,

2016b). It is found that their effects on the absorption are minor compared with those of the overall geometry or size. Thus, we ignore those minor geometric factors in the aggregates, and assume FA to be perfectly aggregated (point-to-point attach without overlapping) by same-sized spherical monomers.

Immediately after emitted into the atmosphere, BC aggregates exhibit lacy structures with a small fractal dimension $D_f$, normally less than 2 (Sorenson, 2001; Chakrabarty et al., 2009). We refer to these lacy-structured BC aggregates as Fresh BC here. Over time, the structure and chemical composition of BC particles change, i.e. a process called "aging". Aged BC particles normally have structures of compact aggregates coated by other material (Moffet and Prather, 2009). We refer to these particles as Coated BC. In some cases (such as humidified biomass burning aerosols), aged BC particles have
structures of compact aggregates without coating (Lewis et al., 2009). We refer to these uncoated aged particles as Compact BC. These three BC geometries are considered in the present study: lacy aggregates, compact aggregates and aggregates with coating, and are referred as the Fresh BC, Compact BC and Coated BC.

The optical properties of the mixed BC were investigated by a core-shell model with a Mie theory in the past, which assumes
a spherical BC core in the center of coating sphere (Chung et al., 2012, Lack et al., 2012; Peng et al., 2016; Moffet and Prather, 2009), and this may introduce significant differences on BC optical properties. Meanwhile, some studies introduced more complex and realistic geometries to consider the effects of coating on BC optical properties (Liu et al., 2012; Dong et al., 2015; Liu et al., 2016b), whereas the AAE was not explored. For Coated BC, this study uses a compact aggregate as the BC core, and a spherical coating is added as the coating material following the numerical model developed by Liu et al.
(2017a). The coating is assumed to be non-absorptive sulfate, the wavelength-dependent refractive indices of which are obtained from the well-known aerosol optical property database OPAC (Hess et al., 1998). The real part of sulfate refractive index decreases slightly from 1.47 to 1.42 as the wavelength increases from 0.3 to 1.0 µm.

However, the amount of coating, the most important factor to determine the absorption enhancement, is one of the most
poorly investigated issues for coated BC (Liu et al.; 2017b). The key issue here is to develop a relationship between core size and coating amount for a group of different-sized BC particles. In other words, after fresh BCs get coated over a certain amount of time, do small cores tend to have the same coating amount as big cores? Or do small cores have larger coating fractions than those of big cores? As pointed out in Section 1, if coating volume fraction is assumed to be fixed for different core sizes, modelled AAE variation is different from observations (Lack and Cappa, 2010). On the other hand, if we apply
aerosol condensation physics, we speculate that small core particles are associated with larger coating amount, since condensation reduces the diameter differences between big and small particles. Thus, a more realistic and observational based relationship between BC core size and coating amount should be derived.

We choose to use the experiment results by Schnaiter et al. (2005), because they gave the size distributions of the fresh and coated BC in a closed chamber environment. They coated diesel soot particles with secondary organic compounds produced by in situ ozonolysis of α-pinene in a large aerosol chamber facility, and BC particles were aged for 24 hours. Figure 5 of their paper shows how the BC size increases as coating progresses. Lognormal size distributions were fitted based on the experimental data, and, during the 24 hours aging process, the geometric mean diameter (GMD) was observed to increase from approximately 0.2 to 0.4 µm, demonstrating the effect of coating. Meanwhile, the geometric standard deviation (GSD) decreased from approximately 1.7 to 1.2. The pair of BC size distributions before and after 24 hours of coating is shown in the Fig. 2a. However, Schnaiter et al. (2005) used a Scanning Mobility Particle Sizer (SMPS) to measure particle size, so the measured size is the mobility diameter ($D_m$). For thickly coated BC (i.e., BC after 24 hours of coating), it can be safely assumed that the mobility diameter equals the diameter of an equivalent volume sphere ($D_{equ}$), because thickly coated BC particles are highly compact and almost spherical. Thus, the size distribution measured by the SMPS after 24 hours of coating can directly represent the equivalent volume size distribution of Coated BC (i.e. $D_{equ}$ is following a lognormal distribution with a GMD of 0.4 µm and a GSD of 1.2).

By assuming that, during the 24 hours of coating, there was no particle coalescence or coagulation, the size distribution of Fresh BC can be taken to represent that of BC cores of Coated BC. For Fresh BC, however, the mobility diameter deviates substantially from the diameter of equivalent volume sphere due to the lacy aggregation structures, and the latter is what we actually need in order to derive the BC core size distribution. To estimate the equivalent volume diameter of BC core (i.e. fresh BC), we first convert the mobility diameter of Fresh BC aggregates to monomer number, and this is achieved by applying Equations (5) and (21) in Naumann (2003), which use the fractal geometric parameters of diesel soot aggregates (Schnaiter et al., 2003). Then, the equivalent volume diameter can be obtained by assuming the $D_{equ} = 2a N^{-3}$ relation, and Fig. 2b illustrates the resulting relationship between $D_m$ and $D_{equ}$ of BC core, i.e. Fresh BC. As expected, $D_{equ}$ of Fresh BC becomes much smaller than $D_m$ as the particles getting larger. Finally, the size distributions of $D_{equ}$ for BC core and Coated BC are both obtained, and they are illustrated in Fig. 3c.

In addition, we are applying the following two simplifying assumptions to map the core size distribution into the size distribution of the Coated BC. (1) Large BC cores are still larger than small BC cores after coating, i.e. BC cores from the left or right wing of the core size distribution will appear on the left or right wing of the Coated BC distribution, respectively. (2) For each singe BC core size there exists only one single coating amount, i.e. applying a "one-to-one" mapping. With those two assumptions, the resulting equivalent volume diameter distribution of the BC core ($D_{equ, BC\ core}$) is mapped to that of the Coated BC ($D_{equ, Coated\ BC}$), and the core size dependent coating amount can be calculated. As a result of this procedure, it is straightforward to estimate the BC core volume fraction in the Coated BC particles: $V_{BC\ core}/V_{Coated\_BC} = (D_{equ, BC\ core}/D_{equ, Coated\_BC})^3$, and the volume fractions of BC core and coating as a function of $D_{equ, BC\ core}$ are visualized in Fig. 2d. As Fig. 2d shows, small BC cores sized in 30~50 nm in equivalent volume sphere diameter account for less than 1% of the total volume, while large BC cores sized in 300~330 nm explain about 10% of the total coated BC particle volume. The relationship between BC core size and the coating volume fraction agrees with the trend obtained by Fierce et al. (2016), who used a

particle-resolved numerical model. The derived relationship between BC core size and the coating volume fraction, as shown in Fig. 2d, is the first experimental-based one to our knowledge, and as such we view it as realistic. Please note that the Coated BC considered in this study (i.e., coating after 24 hours) represents a case study (resembling the findings presented by Schnaiter et al. (2005)) to give insights into the effects of coating on BC AAE. In real atmosphere, aged BC particles have

extremely different and complex shapes and coating amounts (Liu et al., 2017b), both of which may significantly influence the optical properties as well as the AAE.

Figure 3 visualizes some examples of BC particles to illustrate the three BC formats, i.e., geometries considered in this study. A tunable particle-cluster aggregation algorithm is applied to generate the FAs (Filippov et al., 2000; Liu et al., 2012), and the coating sphere is added with its center located at the mass center of compact FA. With the coating fraction as discussed

earlier, coating can totally embed the compact aggregates. Three transmission or scanning electron microscope images of BC particles are also given in the figure for comparison (Burr et al., 2012; Lewis et al., 2009; Freney et al., 2010), and we can see that the numerically generated particles have similar structures to those observed ones. Geometric parameters for FA with observational bases have been widely used for numerical studies (Liu and Mishchenko, 2005; Smith and Grainger, 2014; Li et al., 2016), and are also applied here to represent realistic BC particles. Specifically, the diameter of each monomer is

set to 30 nm as supported by observations (Brasil et al., 2000; Chakrabarty et al., 2014), and the fractal prefactor of 1.2 that was estimated by Sorensen and Roberts (1997) is used. For the lacy aggregate, a fractal dimension of 1.8, close to the is observed average by Köylü et al. (1995) and Sorensen and Roberts (1997) and both the Compact and Coated BC particles use a fractal dimension of 2.8 to make the BC particles as compact as we can. With the coating fraction as a function of the BC core size as in Fig. 2d, the aggregates are completely wrapped inside the coating sphere. Nevertheless, to better

understand the sensitivity of BC AAE to geometries parameters, we will also explore other monomer diameters (i.e. 20 nm and 40 nm) and another fractal dimension of 2.3 (i.e., the mean value between the two extreme values and values derived from observation of Wang et al. (2017)).

## 2.2 BC size distribution

Particle size distribution is one of the most commonly measured variables for aerosol studies. Aerosol size measuring

instruments (such as the SMPS mentioned above, and the single particle soot photometer, i.e. SP2, and electron microscopy) have repeatedly shown that a lognormal size distribution is a good fit for realistic BC size distributions (Bond et al., 2002; Schnaiter et al., 2005; Chakrabarty et al., 2006; Kirchstetter and Novakov, 2007; Reddington et al., 2013; Wang et al., 2015), and it is also widely used in numerical calculations of BC radiative properties and forcing (Moffet and Prather, 2009; Chung et al., 2012; Li et al., 2016). It should be noted that various different quantities are used to describe BC size distribution

according to the principle used for the measurement. For example, as noted before, the SMPS measures the mobility equivalent diameters $D_m$, which is quite sensitive to the particle shape, the SP2 gives the mass equivalent diameter of BC and BC cores (Reddington et al., 2013; Wang et al., 2015), and the projected areas deduced from BC micrographs are also used to understand its size distributions (Chakrabarty et al., 2006). Overall, BC diameters from a few tens to almost 1000 nm are

obtained (Schnaiter et al., 2005; Chakrabarty et al., 2006; Kirchstetter and Novakov, 2007; Reddington et al., 2013; Wang et al., 2015).

To specify and unify the definition, all sizes in this study are referred to as the diameter of equivalent volume sphere from here on. For Fresh or Compact BC, this diameter can be given as $2a\sqrt[3]{N}$, and, to ensure conservation of both mass and size distribution for comparison among Fresh, Compact and Coated BC, the size of Coated BC will also be defined as that of the BC core. With the relationship between the coating fraction and BC core size given in Fig. 2d, it is straightforward to derive the overall size of Coated BC with the diameter of bare BC part known. A median size and a standard deviation are used to describe the lognormal size distribution and to obtain the bulk absorption of an ensemble of BC particles. The GMDs between 0.10 and 0.12 µm are most widely observed and used for numerical study (Alexander et al., 2008; Coz and Leck, 2011; Reddington et al., 2013; Wang et al., 2015), and the GSD values vary within a relatively narrow range. As the advantage of numerical study, we consider a relatively wide range of particle size distributions with the GMDs between 0.05 and 0.20 µm for sensitivity purposes, and a fixed GSD of 1.5 is assumed. For BC core with GMDs between 0.05 and 0.20 µm and a GSD of 1.5, the corresponding Coated BC with the aforementioned coating faction has overall GMDs between 0.15 and 0.28 µm and GSDs of approximately 1.2.

For aggregates with fixed monomer sizes (diameter of 30 nm without special mention, and 20 and 40 nm used for sensitivity studies), their diameters are only determined by the number of monomers in the aggregate. We consider aggregates with the number of monomers ranging from 1 to 2000, corresponding to diameters of equivalent volume spheres from 0.03 to almost 0.4 µm. For example, with a monomer diameter of 30 nm, an aggregate with approximately 300 monomers corresponds to an equivalent-volume sphere with a diameter of 0.2 µm. Numerical integrations can be easily carried out to obtain the bulk absorption of an ensemble of aggregates with a given size distribution. Again, for Coated BC, the size distribution is applied for only the BC core, which keeps conservations on the BC amount and BC size distribution, and the overall effective volume diameter of coated BC is larger than that of pure BC.

## 2.3 BC refractive index

The refractive index (RI), a wavelength dependent complex variable, is one of the most important parameters to determine aerosol AAE, because the absorptions at different wavelengths are significantly influenced by both the real and imaginary parts of RI. However, it is also one of the most uncertain physical properties of BC particles, because it cannot be directly observed. Estimates of BC RI have been normally made from observed absorption, scattering (or extinction) and size distribution of suspended particles, or from reflectance measurements on compressed BC pellets, and the RI is inferred by obtaining a best fit to numerical simulations (either Mie theory by assuming spherical particle shape or the simple Rayleigh-Debye-Gans theory) (Chang and Charalampopoulos, 1990; Schnaiter et al., 2003; 2005; Kirchstetter et al., 2004; Dalzell and

Sarofim, 1969; Stagg and Charalampopoulos, 1993; Vanhulle et al., 2002; Moteki et al., 2010). Some of those studies extend RIs at particular wavelengths into the whole spectrum by the dispersion equations or the Kramers-Kronig analysis (Dalzell and Sarofim, 1969; Querry, 1987; Chang and Charalampopoulos, 1990). These retrieval methods based on the unrealistic spherical shape assumption or inaccurate numerical modeling pose sizable errors on estimated RIs, let alone the error in

5    aerosol optical property measurements. Furthermore, the BC materials from different combustions probably have different RIs, and this was discussed in the past (e.g., Sorensen, 2001; Bond and Bergstrom, 2006). After development over almost half a century, there are also numerous datasets available with BC RIs over the entire solar spectrum to obtain its optical properties for radiative applications related to BC (d'Almeida et al., 1991; Krekov et al., 1993; Hess et al., 1998). More details on the BC RIs have been carefully reviewed and summarized by Sorensen (2001) and Bond and Bergstrom (2006).

Figure 4 compares BC RIs from those cited studies. Most observational based studies give RIs at some specific wavelengths, at which the observations are carried out, and some fitted results with continuous variations are also given. Both real and imaginary parts of the BC particles show quite wide ranges of variations, and we eliminated results with real part much larger than 2 and imaginary part much smaller than 0.5. The real part generally varies between 1.5 and 2.0. The imaginary

15   part shows similar range of variation, and values from 0.5 up to 1.1 have been retrieved. Furthermore, none of these datasets show a wavelength-independent RI, and quite different variations over the wavelength are shown in the figure. The real part of RI generally increases as the wavelength becoming larger, whereas the slopes of the variations are quite different. However, both increasing and decreasing trends are noticed for the imaginary part of RI. The figure clearly shows the uncertainties and large variations on BC RI, which brings the most significant challenge on approximating its AAE.

In view of Fig. 4, it is difficult to find a single value to represent BC RI, whereas it is doable to give a reasonable range of variation for numerical investigation. Considering the significant uncertainty in estimated BC RI due to differences on combustion fuels and conditions as well as whether BC is fresh or aged, we consider both wavelength-independent (i.e. constant) and wavelength-dependent RIs, and introduce two parameters to indicate the variation of real and imaginary part as

25   wavelength, respectively. The real and imaginary part are defined as:

$$Re(\lambda) = Re_o + A(\lambda - 0.55) \tag{4}$$

$$Im(\lambda) = Im_o \times 10^{B(\lambda - 0.55)} \tag{5}$$

where $\lambda$ denotes wavelength in units of micron. A and B represent the wavelength dependence, and are defined independently. We use 0.55 μm as the reference wavelength, and $Re_o$ and $Im_o$ can be understood as the real and imaginary

30   part of RI at 0.55 μm. If A=B=0, the BC RI becomes wavelength independent. The real part of RI is simply assumed to vary linearly with wavelength, and the imaginary part is linear in the logarithmic scale. Because the range of wavelength we consider is relatively narrow, i.e. visible and near infrared range, the simple assumption can capture the general variation of BC RIs. The corresponding A and B values for the RI shown in Fig. 4 are listed in Table 1, and the results from the various datasets show quite different values. It should be noticed that Schnaiter et al. (2003; 2005) gives A and B much larger than

those of other datasets. Besides Schnaiter et al. (2003; 2005), A and B values in the range of (0.0,0.25) and (-0.25, 0.0) respectively are enough to account for the RI wavelength-dependence of other datasets. In light of the RI estimates made by previous studies, both A and B are assigned to be in the range of -0.5 and 0.5 here, which gives a much wider range of variation than those shown in Fig. 4 (besides those from Schnaiter et al. (2003; 2005)). The shadow areas in Fig. 3 represent

the large range of RIs that is considered in this study, and the areas clearly cover most previous BC RI estimates. It should be noticed that the real and imaginary parts of RI don't change independently (Bond et al., 2007; Moteki et al., 2010), and A and B can only change with the limitations of the dispersion equations or the Kramers-Kronig analysis. However, we want to take the advantage of numerical models to better understand the effects of each parameter on BC AAE, and, thus, A and B are assumed here to be independent. This may lead to an overestimation of the AAE variability due to non-realistic RI.

**3 BC AAE**

With BC shape, size distribution and RI known, it becomes straightforward to calculate the corresponding optical properties at a given wavelength, and we only consider bulk properties averaged over a given size distribution in this study. Multiple numerical models are available to account for the light scattering properties of a cluster of spheres, where the individual spheres of the cluster do not overlapping, and the Multiple-Sphere T-matrix Model (MSTD) developed by Mackowski and

Mishchenko (2011) is used in this study. The MSTM is a numerically exact model for light scattering by multiple spheres, and is widely used to study the scattering properties of BC particles. Due to the high accuracy and efficiency provided by the MSTM, it becomes convenient to consider the optical properties of BC as aggregates of small spherical monomers. In the framework of the MSTM, the BC particles are rigorously treated as FAs shown in Fig. 3 for optical property simulations, so that the errors can only be introduced by uncertainties in the particle microphysical properties, and not the numerical model.

Furthermore, the MSTM is also capable of considering the interaction among a large sphere and small ones embedded inside the host, which is also the exact configuration for the Coated BC case in this study.

The AAE is widely approximated with the absorptions at two wavelengths using Eq. (2) (Utry et al., 2014; Li et al., 2016), and BC shows noteable different AAE values over different ranges of the wavelength spectrum. To obtain the most

representative AAE value, we use BC absorptions cross sections at multiple wavelengths between 0.3 and 1.0 μm in steps of 0.05 μm, and the best AAE value to fit these cross sections over the spectrum is obtained by a linear regression of the log-transformed data (i.e., between $ln(C_{abs}(\lambda))$ and $ln(\lambda)$ in the logarithm format of Eq. 1). Figure 5 illustrates an example of the AAE calculation, in which the averaged bulk absorption cross sections are shown in the logarithmic scale as a function of wavelength. The red crosses in the figure are obtained from the MSTM for the Fresh BC particles integrated over the BC

lognormal size distribution (with a GMD of 0.12 μm and a GSD of 1.5). A wavelength-independent RI of 1.8+0.6$i$ over the entire spectral range are used for the simulation. The absorption cross section shows a clear linear variation in the logarithmic coordinate, ad indicated by the blue line, which represents the result of a linear regression fit. Thus, the slope of

the line, i.e. 1.04 in this figure, is the AAE for BC with the corresponding microphysical properties. In this way, the bias introduced by considering absorptions at only two wavelengths can be avoided. It should be noted that the example in the figure shows an excellent linear relationship, which is not true for all cases, whereas linear regression still gives the best representation on the AAE values if the absorption doesn't accurately decrease exponentially.

Figure 6 shows the calculated BC AAE with a wavelength-independent RI of 1.8+0.6$i$ using the aforementioned numerical methods, and the sensitivities of BC AAE to particle geometry and size are clearly illustrated. The results for spheres with equivalent volume are also given in the figure, and show obviously different variation from those of non-spherical particles, which demonstrates the necessity to consider the complex geometries of BC particles. Different BC conditions in the atmosphere, i.e. Fresh, Compact, and Coated BC, are represented by particles with different geometries. As evident in Fig. 6, even for these relatively small aerosols, geometry plays an important influence on the AAE. Moreover, particles with different geometries show different dependences on particle size. Liu et al. (2015b) pointed out that the AAE could also sensitive to monomer size, which is fixed to have a radius of 15 nm in our work. Thus, the shade in Fig. 6 depicts the influences associated with different monomer radii, i.e. a=10 nm and a=20 nm, and all other results shown in this study use a radius of 15 nm supported by observations. Monomer size is found to be not very impactful on the AAE of lacy aggregates, whereas the AAE may decrease by as much as 0.1 or more for large Compact BC as the monomer radius increases from 10 to 20 nm. The AAE of the Fresh BC is approximately 1.05 and insensitive to particle size, because the interaction between monomers for lacy aggregates are relatively weak and the absorption per monomer does not change significantly as BC aggregate becomes larger. However, based on the Rayleigh theory, the AAE of small BC with a wavelength-independent RI should be 1.0 (Moosmüller and Arnott, 2009), because the Rayleigh absorption for small particles is proportional to $\lambda^{-1}$. Because the Rayleigh approximation underestimates particle absorption for relatively small aggregates as wavelength becomes smaller (see Fig. 1 and the corresponding discussion), especially those with aggregation structures, and, thus, accurate scattering simulations give AAE values a little bit larger than 1.0 for even small-sized particles with wavelength-independent RI. Meanwhile, the AAE of the Compact and Coated BC is highly sensitive to particle size, and decreases sharply with the GMD. The Compact BC has smaller AAE than Fresh BC, and shows an AAE as low as almost 0.8 as the GMD becomes close to 0.2 μm. For aggregates with a fractal dimension of 2.3 (Wang et al., 2017), the AAE curve lies right between those of Fresh ($D_f$=1.8) and Compact ($D_f$=2.8) BC, and this further demonstrates the clear influence of aggregates structure on BC AAE. The AAE of Coated BC is even more sensitive to particle size, decreasing from almost 1.4 to 0.8. For Coated BC with GMD (again, this refers to the GMD of the BC core, not the total inhomogeneous particle) larger than approximately 0.10 μm, coating would decrease the AAE of Fresh BC, and the Coated BC gives AAE values comparable (slightly smaller) to those of Compact BC with GMD larger than 0.14 μm. Meanwhile, both those relatively small AAE values obtained for Compact and Coated BCs can potentially explain observed small BC AAEs (Kirchstetter et al., 2004; Arnott et al., 2005; Gyawali et al.; 2012; Chakrabary et al., 2013). The AAE of Fresh BC with a GMD of 0.2 μm drops from approximately 1.05 to 0.8 after aging, both of which are close to the observed values and coincident with the changes (from

1.1 to about 0.8) of Diesel soot from Schnaiter et al. (2005), and the differences may be caused by uncertainties on the RI or coating amount. Even with non-absorptive coating material, the coating still amplifies BC absorption and changes the BC AAE (Liu et al., 2017a). Considering the aging processes of BC aggregates, Figure 6 shows that, for relatively large BC particles, both Compact and Coated BCs have smaller AAE than those of the Fresh BC with lacy particle structure. The coating amount, structure or material may affect the absorption enhancement at different particle sizes and then the AAE, and this study considers a special but observation-based case. As a result, the conclusion for the Coated BC should be more carefully tested for further applications.

The influences of particle RIs on the AAE are illustrated in Fig. 7, and results for the Fresh, Compact and Coated BC are shown from top to bottom panels. Figure 7 considers only wavelength-independent RIs. The left column is for BC particles with a fixed imaginary part of 0.6 but real parts of 1.6, 1.8 and 2.0, and the right column is for those with the same real part (1.8) but different imaginary parts (0.4, 0.6, 0.8 and 1.0). The BC AAE increases as the real part increases or the imaginary part decreases. Although the imaginary part of RI is most directly related to particle absorption, real and imaginary parts affect BC AAE to similar degrees. Again, the sensitivity of BC AAE to its RIs is quite different for particles with different geometries. The AAEs of the Fresh BC is less sensitive to RI, and shows difference of < 0.1 for the RIs we considered. As the RI real part increases from 1.6 to 2.0, the AAE of the Compact BC increases by approximately 0.15, and the changes reach to as large as 0.3 for BC imaginary part between 0.4 and 1.0. However, after coating, the AAE becomes less sensitive to RI real part, but more sensitive to RI imaginary part. This means that, with the BC core totally embedded by the non-absorptive coating, the absorption enhancement of Coated BC is more sensitive to the IM of BC RI.

All previous studies used a wavelength-independent RI over the entire spectrum, which may or may not be realistic for BC particles in the atmosphere. As explained in Section 2.3, parameters A and B are introduced to account for wavelength variance of the real and imaginary parts of RI, respectively. Because BC absorption increases as the real part of RI decreases or the imaginary part increases, it is simple to understand the influence of wavelength-dependent RIs on BC AAE. With simple variations assumed for both RI real and imaginary parts, it becomes possible to quantify the effects of RI wavelength variations. Previous simulations have given BC absorptions at some particular RI values (i.e. real part of 1.6, 1.8, and 2.0, and imaginary part of 0.4, 0.6, 0.8 and 1.0), and, to save computational time, BC absorptions at other RIs (that are obtained from Eqs. (4) and (5) with given A and B) are approximated by interpolation among those existing results. By comparing with accurate MSTD results, we find that the interpolation introduces relative errors of < 1% for the absorption, which is accurate enough for the AAE simulation.

Figure 8 illustrates the impacts of wavelength-dependent RIs on BC AAE, and three rows correspond to results for the Fresh (top), Compact (middle) and Coated (bottom) BC. Both A and B are assigned to change between -0.5 and 0.5. Figure 8 considers two examples for the reference RI at the wavelength of 0.55 μm, i.e. $Re_o + Im_o i$ (1.8+0.6$i$ and 1.8+0.8$i$), and they

are marked by the solid and dashed curves, respectively. Meanwhile, three BC equivalent-volume diameter distributions with the GMDs of 0.05 μm, 0.12 μm and 0.20 μm are used, and visualized by the red, blue and green curves. As expected, the AAE increases as A increases or B decreases, and quite different slopes are shown in different panels. Quantitatively, the AAE of BC with wavelength-dependent RIs for typical values of A=0.2 and B=-0.1 (Chang and Charalampopoulos, 1990;

Krekov, 1993) would be approximately 0.15 larger than those with a wavelength-independent RI, and it becomes more difficult to understand the observed small BC AAE values (Kirchstetter et al., 2004; Gyawali et al., 2012). We focus on two features illustrated by the figure, both of which will be used later to decompose the influences of different BC properties on its AAE. First, for all cases, the AAEs show relatively linear variations as either A or B changes, and, thus, it becomes easy to quantify the influence of wavelength-dependent RIs on BC AAE. Secondly, different panels show very different AAE

variation (slope) over A or B, whereas, for each panel, the slopes for different cases (i.e. different curves in a panel) are relatively similar, which means that the influence of both A and B dependent only on particle geometry. Figure 8 only shows six special but typical cases (two $Re_o+Im_o i$ by three GMDs), and more tests were carried out, which show very similar results. Again, it should be noted that, due to the neglect of the dependence on A and B values, these results may overestimate the variability on AAE to some degree.

With all previous factors considered, it becomes possible to decompose the influences of BC properties on its AAE. The BC properties show clearly monotonic influence on the AAE but to varying degrees. BC morphology, the most complex property parameterized by multiple parameters (e.g., fractal parameters, coating parameters and coating amount) can hardly represented by a single variable, whereas plays the most important and complex role in determining BC AAE. Thus, the

morphology is considered independently, and its influence can be considered with other parameters known. To be more specific, the relationships between the BC AAE and the properties besides shape are simply approximated with linear equations for each BC geometry, and, thus, the BC AAE is expressed by:

$$AAE = AAE_o + k_1 log \frac{GMD}{0.12} + k_2(Re_o - 1.8) + k_3(Im_o - 0.6) \qquad (6)$$

where GMD is given in unit of μm, and 0.12 μm can be understood as the reference size. Considering the AAE variation

over different variable, a logarithmic relationship is used for the BC size, and all other relationships are assumed to be linear. The coefficients, from $k_1$ to $k_3$, are fitted to indicate the significance of the corresponding influence on BC AAE. The GMD should be given in unit of micron. Because the influence of BC properties on its AAE for particles with different geometries are completely different, we approximate the coefficients of the above equation for the Fresh, Compact and Coated BC separately. The fitted coefficients are given in Table 2. The $AAE_o$ and three coefficients ($k_1$, $k_2$, and $k_3$) in the table are

obtained by given the smallest root-mean-square relative errors for all AAE values calculated based on wavelength-independent RIs. The influence of BC properties on its AAE are clearly demonstrated by the coefficients in Table 2. Large absolute values mean to have more significant influence on the AAE, and the positive or negative sign indicates the sign of the correlation. The effects of RI wavelength dependence (i.e., A and B in Equations (4) and (5)), which have been shown in

Figure 8, are not included in Equation (6), because they are less quantitatively meaningful due to the significant uncertainties on the dependence between A and B.

To demonstrate the performance of the simple expresses on approximating BC AAE, Figure 9 compares BC AAEs from accurate absorption simulations and Eq. (6). The left panel of the figure shows the cases in which the approximations give relatively accurate agreement to the simulations, whereas some of the poor cases are illustrated in the right panel. It is clear that the results for the Fresh BC show the best agreement because of the relatively weak influence of particle size on the AAE. The relatively poor agreements for the Compact and Coated BC are mainly because of the non-linear variation of the AAE on the GMD. The differences between the accurate simulations (solid curved) and the approximation (dashed lines) can reach slightly over 0.1 for small BC particles. However, considering that the BC are widely observed or considered to have a GMD larger than 0.10 μm, Figure 9 indicates that the simple linear approximation we obtained give a quite accurate estimation on BC AAE for BC of the three types.

Our results obtained here indicate that BC microphysical properties have a clear influence on BC AAE, and the influence can be quantitatively understood by Eq. (6) and the coefficients listed in Table 2. Considering the obvious variations obtained for BC particle sizes and geometries and the uncertainties on its RI, it is impossible to find a "best" AAE value for all BC aerosols. However, a range of reasonable AAE values can be obtained based on the observations of BC properties and the numerical results. For Fresh BC, the AAE are approximately 1.05, and aging makes AAE of typically sized BC particles to decreases by approximately 0.15 to 0.90. Furthermore, wavelength-dependent RIs makes the case much more complicated, and give wider ranges on the BC AAE. For Coated BC particles, the absorption can also be significantly affected by chemical composition, amount and geometry of mixing material (Li et al., 2016), and this study introduces a laboratory-based coating amount distribution to reveal the significant effects of coating on BC AAE. With the help of Eq. (6), the BC AAE can be easily calculated if its properties (shape, size and RI) are known.

## 4 Summary and conclusions

We have numerically investigated the AAE of BC aerosols of three states in the atmosphere, i.e., fresh, compact and coated ones. The numerical computations conducted here have multiple controllable variables (such as BC size distribution) that all effect BC AAE, and we attempted to constrain these variables within the realistic ranges as determined by observational based studies. The MSTM was used to accurately compute the light absorption of non-spherical particles, and the numerical results were analyzed to better understand the BC AAE values in relation to the controllable variables.

The results challenge conventional beliefs. With a wavelength-independent refractive index, our accurate numerical results show typical BC AAE values of 1.05 and 0.90, instead of 1.0, for fresh and aged BC particles respectively. In reality as

revealed by many observational studies, the BC refractive index likely has sizable wavelength dependence, and BC is often coated by non-BC aerosol materials. In these cases, BC AAE can even move beyond a range of 0.5~1.5. As a result, using a flat value of 1.0 for BC AAE could very likely introduce significant errors in aerosol absorption analysis studies.

Our results demonstrate that BC particle shape is the most influential factor in determining BC AAE. The AAE of fresh BC in the form of lacy aggregate is less sensitive to particle size, whereas, after aging processes, the AAE of BC with compact or coated structures may significantly decreases as particle size increases. As the most uncertain particle property, the refractive index cannot be directly measured, and, thus, brings the most significant challenge on determining BC AAE. To quantify the complicated influences of different BC parameters on its AAE, linear approximations for BC AAEs in different
conditions were obtained here. Our results clearly demonstrate the importance of various parameters on the BC AAE and the errors of assuming BC AAE as 1.0. However, it should take caution to interpret our results as a comprehensive guide or absolute reference, because the closure studies between numerical models and observations on BC properties can be relative poor (Bond and Bergstrom, 2006; Radney et al., 2014).

**Acknowledgements**

We are deeply thankful to D. W. Mackowski and M. I. Mishchekno for the MSTM code. The authors also gratefully acknowledge the effort by the two anonymous reviewers and Dr. Joel C. Corbin to improve the manuscript. This work was financially supported by the National Key Research and Development Program of China (2016YFA0602003), the Natural Science Foundation of China (41505018), the Natural Science Foundation of Jiangsu Province (BK20150899), the Young Elite Scientists Sponsorship Program by CAST (2017QNRC001), and the US NSF (Award No. AGS-1455759). The
computation of this study was supported by the National Supercomputer Center in Guangzhou (NSCC-GZ).

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

**Table 1.** Fitted values for the parameters A and B used to define the spectral variation of BC refractive indices.

| Reference | A | B |
|---|---|---|
| Krekov, 1993 | 0.15 | -0.09 |
| d'Almeida et al., 1991 | 0.01 | -0.05 |
| C and C, 1990 | 0.23 | -0.22 |
| Schnaiter et al., 2003 | 0.63 | 0.36 |
| Schnaiter et al., 2005 | 0.79 | 0.79 |
| Kirchstetter et al., 2004 | | -0.05 |

**Table 2.** Fitted coefficients to show the sensitivities of BC properties on its AAE values. Two significant figures are kept for all coefficients.

| | $AAE_o$ | $k_1$ | $k_2$ | $k_3$ |
|---|---|---|---|---|
| Fresh BC | 1.05 | - 0.04 | 0.24 | -0.13 |
| Compact BC | 0.90 | -0.3 | 0.40 | -0.42 |
| Coated BC | 0.95 | -0.5 | 0.20 | -0.54 |

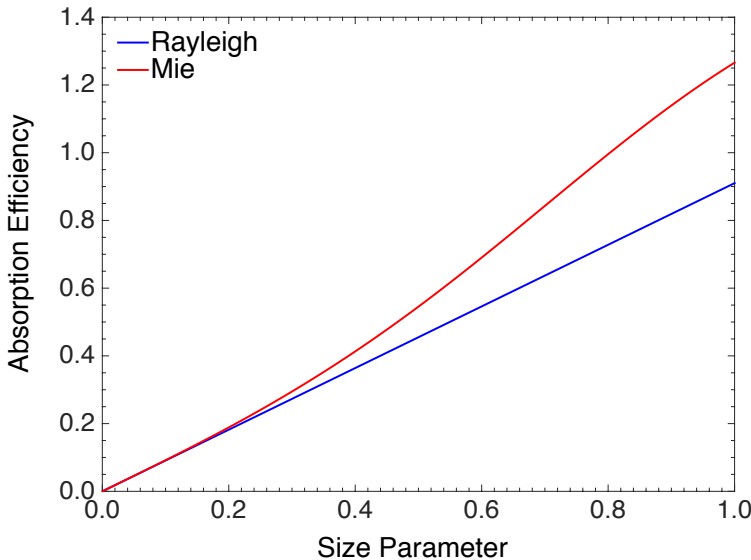

**Figure 1.** Comparison of the Rayleigh approximation and Mie theory for the absorption efficiency of spheres with a refractive index of 1.8+0.6*i*.

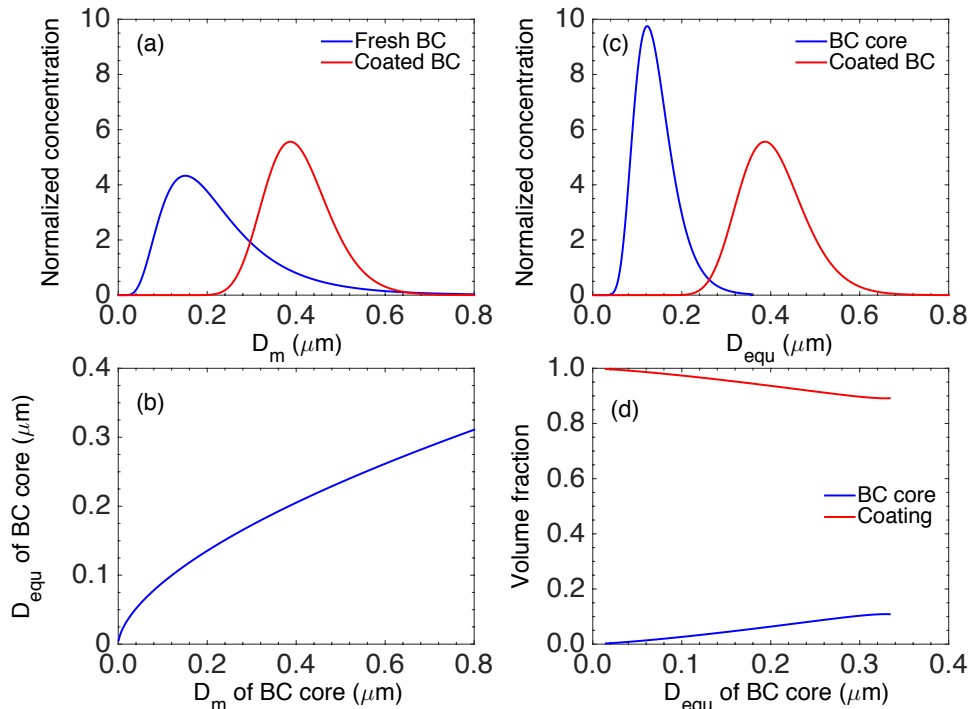

**Figure 2.** (a) Normalized particle number concentration of uncoated (i.e. fresh) and coated (after 24-hour coating) BC, assuming lognormal size distribution and utilizing a laboratory experiment by Schnaiter et al. (2005; see Fig. 5 of their study). (b) Estimated relationship between the diameter of equivalent volume sphere ($D_{equ}$) and mobility diameter ($D_m$) for Fresh BC aggregates. (c) The size distributions for the diameters of the equivalent volume spheres ($D_{equ}$) of BC core and Coated BC after 24-hour coating, based on (a) and (b). (d) Estimated volume fractions of BC core and coating material as a function of $D_{equ}$ of BC core, based on (c).

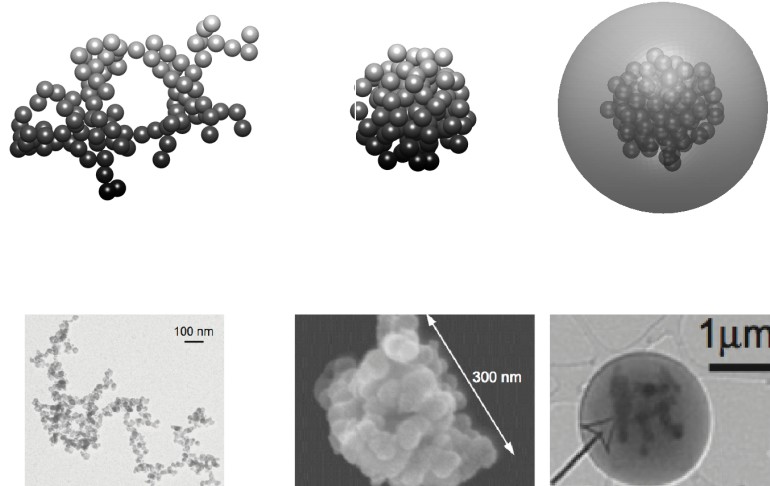

**Figure 3.** Geometries of numerically generated BC aggregates with different particle geometries, i.e. loose aggregate (left) for Fresh BC, compact aggregate (middle) for Compact BC, and coated aggregate (right) for Coated BC, and some examples of realistic BC images for comparison (Burr et al., 2012; Lewis et al., 2009; Freney et al., 2010). Again, the monomer diameters are assumed to be 30 nm. A fractal dimension of 1.8 is assumed for the Fresh BC, and 2.8 is used for the Compact and Coated BCs.

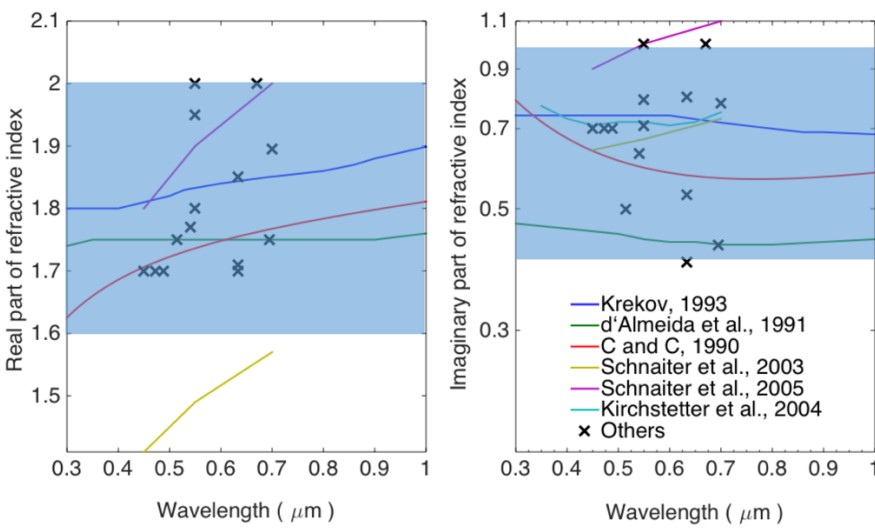

**Figure 4.** The real and imaginary parts of BC refractive indices from various observations. The blue shadow depicts the ranges considered in this study.

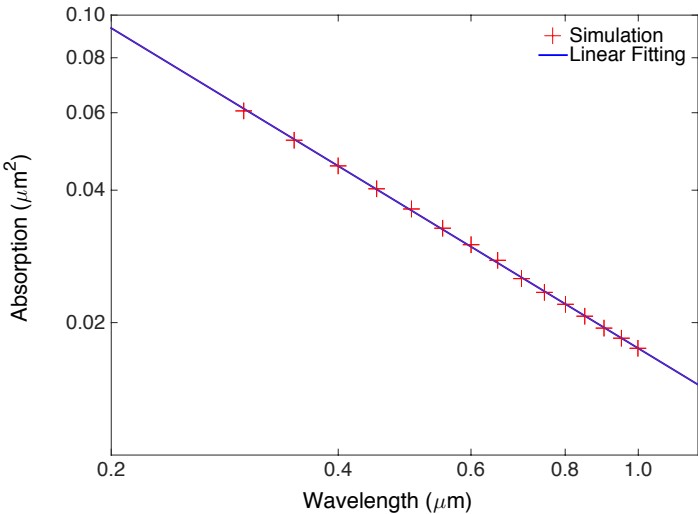

**Figure 5.** Absorption cross sections of Fresh BC with a wavelength-independent refractive index of 1.8+0.6i and a GMD of 0.12 µm as a function of wavelength.

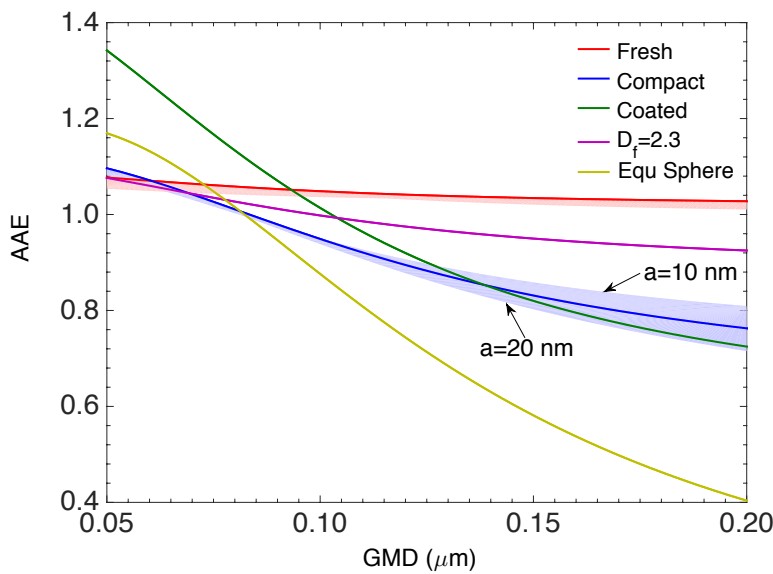

**Figure 6.** AAEs of the Fresh, Compact and Coated BC as a function of the volume equivalent GMD of the BC particles (or BC core for Coated BC).

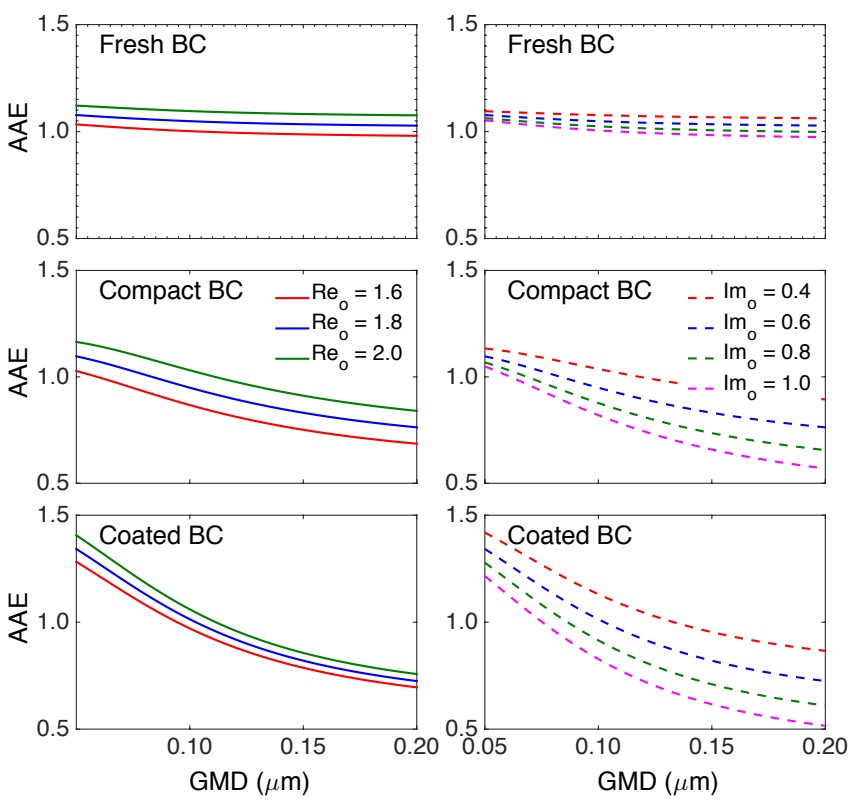

**Figure 7.** Influence of wavelength-independent refractive index on the AAEs of the Fresh BC (top penal), Compact BC (middle panel), and Coated BC (bottom panel). A fixed imaginary part of 0.6 is used for the left panel, and a fixed real part of 1.8 is used for the right panel.

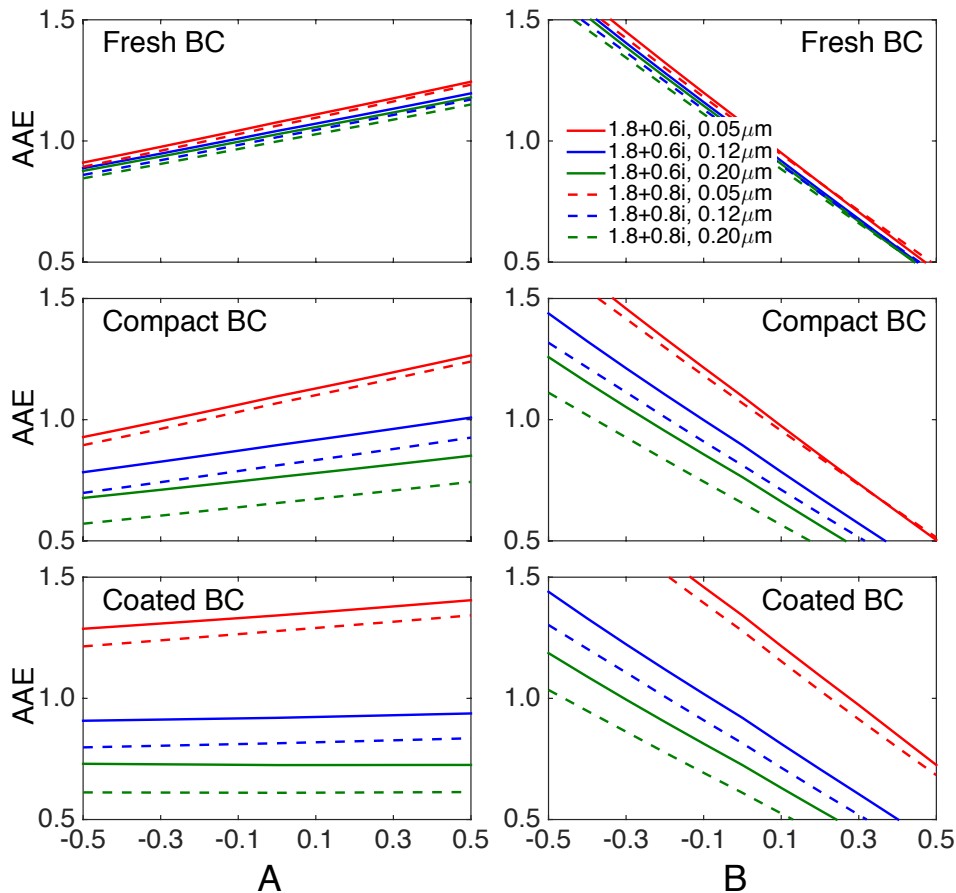

**Figure 8.** Influence of wavelength-dependent refractive indices on the AAEs of the Fresh BC (top penal), Compact BC (middle panel), and Coated BC (bottom panel).

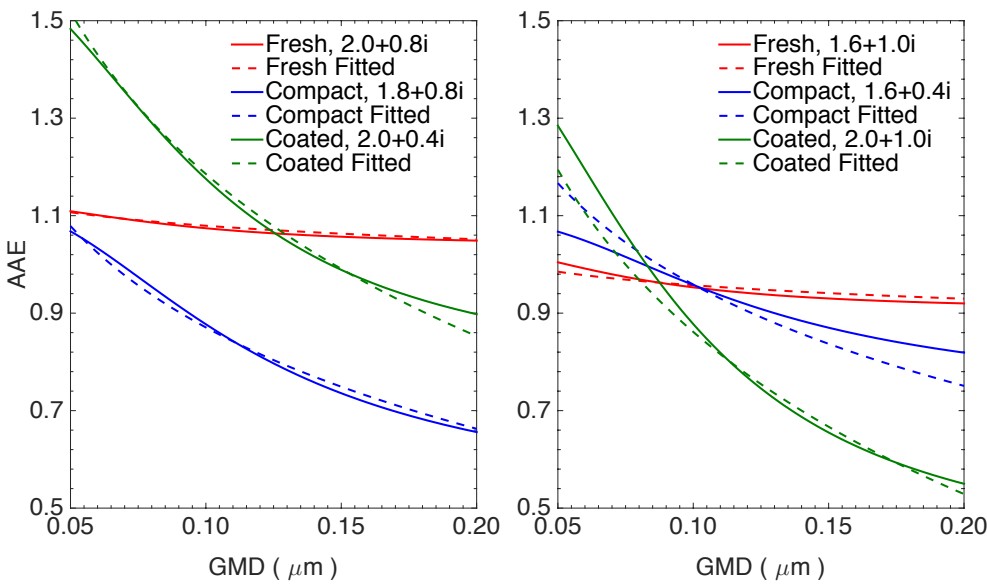

**Figure 9.** Comparison between the AAEs given by accurate numerical simulations (solid curves) and those approximated by Eq. (6) and the corresponding coefficients in Table 2. (dashed curves)

