# Peer review of "The Absorption Ångström Exponent of black carbon: from numerical aspects"

_Atmospheric Chemistry and Physics, 2017_

## Referee Comment (RC1) · Anonymous Referee #1 · 2 Nov 2017

This paper systematically tested the sensitivity of the AAE of BC in three representative morphology, and point out which factors should be considered when deriving AAE from possible available measurements. Though the calculation itself is not new, but the concept and focus is scientifically important. This paper is well organized and generally well written, but in this version it reads a bit too technical, so I would recommend final publication after incorporating a bit more work, to allow this work within the scope of ACP.

Major points:

1) the most lack of this study is the authors have not calculated the AAE of BC in bulk but only for single BC particle. If I understand correctly, the authors have only given the BC lognormal size distribution, coating distribution as a guidance of size range

selection for sensitivity test, however the single BC particle calculation has not been applied in the particle distribution to work out how these calculation will influence the whole. The information in bulk may be more valuable for the ambient measurement as most of the instruments measure in bulk.

2) The particle size as called GMD in this study is a bit confusing. For the coated size, I presume this is the size as entire BC particle, i.e. the coated particle, but if we compare everything all in GMD, would the coated BC has a less content of BC core? I'm not sure how comparable are they if in a same figure. Also, given the BC has complex morphology, what is GMD, is it supposed to be volume-equivalent diameter? This is important to be clarified.

3) How the coating has been associated with BC core is not clearly presented, are they partly coated or embedded? How did you treat the coating interaction with BC? One recent study (Liu et al., 2017, DOI: 10.1038/ngeo2901) could be referenced in page 6 line 10 or page 10 line 28 etc. to support your discussion.

4) The empirical equation (equation 6) is almost all about refractive index uncertainty, and they are separately discussed for three different morphology cases. Though the refractive index has large variation from different literatures, but mostly we are using a fixed refractive index or fixed spectral dependence of refractive index, otherwise there will be no real value for anything. However the authors have not really given how the BC morphology has actually influenced AAE, such as Df value, the amount of coatings associated. These are most interested to communities who care about how the BC ageing will influence its mixing state/morphology and how the AAE will be modified by these factors.

Others: In page 6, the representation of coating thickness according to Schnaiter et al. (2005), could you point out which source of BC are they, and are they fresh BC, how long have they been aged?

Page 6 line 30 to page 7 line 10, there are many parameter assumptions which have

not been clearly explained: the100 monomers are used, so are we actually only resting one BC core size? Have we tested the sensitivity to different monomer sizes (only 30nm is used here)? Liu et al., 2015 (DOI: 10.1002/2014GL062443) point out the AAE could be sensitive to the monomer size, also give the reference you choose 30nm.

For compact BC, the Df is used as 2.8 which is nearly sphere, any reference for this value? As above, it would be useful to test the sensitivity to Df.

Page 7 line 8-16: the whole discussion here is rather confusing, you should point out what size previous instruments actually measured, the coated particle size or only BC core size, currently they are mixed up. You should point out SMPS measured mobility diameter is very sensitive to the particle shape (which is different from the volume equivalent diameter you present here), but references you referred Reddington et al., 2013; Wang et al., 2015 used the BC core size, measured by the single particle soot photometer.

---

## Author Comment (AC1) · 4 Nov 2017

First, we would like to thank the reviewer for his/her valuable comments, and these suggestions will significantly improve the manuscript. We agree with him/her that the topic and focus of this study is quite important, and this study tries to provide a rigorous and systematic research on BC AAE. However, the reviewer may misunderstand some details of this study due to our unclear discussions. We would like to present a short reply at this point, and a more detailed response as well as revision will be provided after the discussion is closed.

1. Actually, all results shown in the manuscript are bulk scattering properties averaged over an ensemble of particles, and, thus, we only mention the integral over particle size

once at Page 7 line 25-27. Because the corresponding discussion is not emphasized enough in the paper (Actually we found that it is only mentioned once briefly), and readers may easily miss the point. Thus, we really thank the reviewer for pointing this out, and we will include much more discussions in the revision to indicate that all results are based on bulk properties.

2. To carry out a fair comparison, the results are illustrated with a constant BC amount/volume. Thus, the GMD is used to describe the diameter of the corresponding volume-equivalent sphere for aggregates, and that of BC core is considered for the coated particles. Furthermore, the MSTD used in this study is an accurate model to account for the interaction between BC aggregates and the coating sphere. We will clarify those points with much more details in the rivision.

3. We did miss some of the key references in the manuscript, and we will include more in the revision. Thanks the reviewer for providing the corresponding comments.

We will stress other comments in details after the discussion is done.

---

## Referee Comment (RC2) · Anonymous Referee #2 · 20 Nov 2017

Inspecting aerosol AAE is important for multitude of applications, and the originally derived values are problematic, as the authors correctly point out.

The authors want to be able to break AAE (a derived particle-scale quantity) into, essentially, microphysical factors contributing to AAE, and this is indeed how I also see the field moving forward. While derived quantities are useful in same situations, BC can be argued to be too varied, either due to coating or differences in aggregation, for a single number to reliably represent all cases.

I think the work done here is important and of high quality, and is definitely worth being published. I have a few main questions I'd like to see addressed as well as a few minor comments.

[Figure]

Major notes:

Is the coating always observed to be a sphere encompassing the aggregate? For a less compact aggregate would the coating be a very large sphere or rather would the shape follow that of the aggregate? Is that something that could be studied with the model used here?

Individual BC particles are not perfect spheres nor perfectly smooth in reality. Is it possible to investigate the effects while taking these non-idealized factors into account? Do the authors think these factors would have significant effects or can they safely be ignored?

Is there a way to parametrize the results by using e.g. equivalent-sized (coated) Mie spheres with some kind of an effective medium approximation? This would greatly help facilitate the use of these types of particles in many applications (moreso than just a better estimate of AAE).

Minor notes:

Page 9, line 16: "straightforward", not "straight forward"

Page 10, line 13: Reference is italicized, unlike most other references. Should be consistent

Fig. 2: X-axis says "Evquivalent", should be "equivalent"

Fig. 9: Legend says "Comapct" instead of "Compact"
* * *

---

## Short Comment (SC1) · 24 Nov 2017

The manuscript "The absorption Angstroem exponent of black carbon from numerical aspects" by Liu, Chung and Yin (ACPD 2017, doi:10.5194/acp-2017-836) is on a very interesting topic. The authors present numerical simulations of BC properties for 3 test cases of morphology using computer-modelled soot aggregates.

The value of these results is clear; the effects of variations in BC microphysical properties on the AAE are clearly seen. However, the limitations of the results could be discussed in much more depth. I have several comments related to this:

1. The words "diameter" and "size" are used in multiple ways throughout the study. Especially when discussing the results of Schnaiter et al., with respect to coating

thicknesses of BC, the word diameter is poorly defined as it would refer only to the apparent mobility of the particles (increased by coatings but decreased by restructuring).

2. The use of a change in mobility diameter (Schaiter et al.) to infer size-dependent coating thicknesses is invalid, as noted above. As a reviewer noted, Liu et al. (Nat. Geosci 2017) should be referenced here. In any case, I am not sure of the value of this size-dependent coating thickness. Since all of the results were plotted against diameter, there is no possibility that the reader misunderstands that coating thickness depends on particle size and history. A constant volume fraction of coatings would allow a fair evaluation of the size dependence to be made. Currently the x axis of the figures (labelled GMD) reflects changes in both size and coating thickness, and is therefore difficult to interpret.

3. The number of primary particles in an aggregate was mentioned as hundreds or even thousands, but what is the most likely number?

4. (page 6) The authors might cite the previous studies which have modelled coatings on complex BC morphologies, eg Liu et al. 2016 (doi:10.1016/j.jqsrt.2015.08.005) and references therein. As the authors noted, however, no previous study discussed the AAE.

5. To what degree might the results change with different choices of monomer diameter and monomer number?

6. On page 11, lines 10-14 do not follow from the previous discussion. The authors are saying that the BC RI may change between sources, so that the AAE may not be fixed. But it has been assumed that the imaginary and real RIs are free to vary independently, which is not a justified assumption (Bond et al. 2006; Moteki et al. JAS 2010, doi:10.1016/j.jaerosci.2010.02.013). This assumption would have

led to overestimated variability in AAE. Also, this assumption was necessary to define A and B as independent in equations 4 and 5.

Also I have a couple of minor comments which were not noted by the reviewers:

- Palas soot (page 3) should be referred to as spark-generated carbon nanoparticles.

- At page 4 line 20, Moosmüller et al. (2011, doi:10.5194/acp-11-1217-2011) should be cited.

- In Section 2.2, please rewrite the discussion. It currently sounds like you are saying that an SMPS measures the diameter of volume equivalent spheres.

- On page 10, I don't understand why a linear regression (I assume of log-transformed data?) should give the best representation of the AAE if the relationship is not exponential. A linear regression of log-transformed data is not reliable even when the data follow a power-law relationship (Clauset et al., 2009, doi:10.1137/070710111)

Finally, the authors have proposed that their numerical results are a suitable absolute reference for BC properties. This is clearly not the case, as a closure between experiment and theory is still lacking (Bond et al., 2006; Radney et al., 2014). Radney et al. (2014, doi:10.1021/es4041804) is particularly relevant to this study because they have used the same T-matrix approach in combination with direct measurements of uncoated BC, yet were unable to satisfactorily model particle extinction. This shows that T-matrix models of BC aggregates should not be used to make strong conclusions on BC properties. Rather than interpret these numerical results as a comprehensive guide and absolute reference, the authors could interpret these results as a clear demonstration of the importance of various parameters on the optical properties of BC. As noted

above, I found these results to be quite interesting, because they demonstrate quantitative the theoretical impacts of changes in various properties on BC in a way that has not been done before.

**References**

All references were either made in the submitted manuscript or were cited using doi's. These doi's provide direct links to the articles via http://dx.doi.org
* * *

---

## Author Comment (AC2) · 26 Dec 2017

Overall: First of all, we would like to thank the two anonymous reviewers and Dr. Corbin for their thoughtful review and valuable comments to the manuscript. In the revision, we have accommodated all the suggested changes into consideration and revised the manuscript accordingly. All changes are highlighted in RED in the revised manuscript.

This paper systematically tested the sensitivity of the AAE of BC in three representative morphology, and point out which factors should be considered when deriving AAE from possible available measurements. Though the calculation itself is not new, but the concept and focus is scientifically important. This paper is well organized and generally well written, but in this version it reads a bit too technical, so I would recommend final

publication after incorporating a bit more work, to allow this work within the scope of ACP.

Response: Thanks for the comments. We agree with the review that the technique may be not new, while the topic and focus of this study is really important. First, we would like to stress that all results were bulk properties averaged over a given particle size distribution. Because the simulations for the bulk properties are not well emphasized in the original version, it may be missed by the readers. Considering that some comments are related to the ensemble average and bulk properties, we improved the discussion to avoid misunderstanding. Meanwhile, all comments are constructive and important to improve the manuscript, and we followed the suggestions to incorporate our work.

Major points:

1) The most lack of this study is the authors have not calculated the AAE of BC in bulk but only for single BC particle. If I understand correctly, the authors have only given the BC lognormal size distribution, coating distribution as a guidance of size range selection for sensitivity test, however the single BC particle calculation has not been applied in the particle distribution to work out how these calculation will influence the whole. The information in bulk may be more valuable for the ambient measurement as most of the instruments measure in bulk.

Response: We totally agree with the reviewer that the bulk properties are more valuable for ambient measurements and downstream applications than that of a single particle, and, actually, we only discuss bulk properties in the work. However, the simulations for the bulk absorption is not well introduced in the original manuscript, and readers may easily miss the point. All results shown in this study are bulk properties averaged over a lognormal size distribution. We considered size distributions with geometric mean dimeters ranging from 50 nm to 200 nm, and a fixed geometric standard deviation of 1.5. To avoid similar misunderstanding, we emphasize the process of the corresponding simulations. (Line 15 of Page 1, Line 3 of Page 9, and Line 25 of Page

10)

2) The particle size as called GMD in this study is a bit confusing. For the coated size, I presume this is the size as entire BC particle, i.e. the coated particle, but if we compare everything all in GMD, would the coated BC has a less content of BC core? I'm not sure how comparable are they if in a same figure. Also, given the BC has complex morphology, what is GMD, is it supposed to be volume-equivalent diameter? This is important to be clarified.

Response: The GMD is one of the two parameters in the lognormal size distribution to determine BC size distribution, and the other parameter, GSD, is fixed to be 1.5. At this point, we want to clarify two points: (1). All GMDs are specified for the BC part to keep the BC amount consistent. Thus, the coated BC will have larger overall sizes than those of bare BC with the same GMD, but the amount and size distribution of BC component are consistent for a fare comparison. (2). For different BC particles with complex morphology, the size is defined by the diameter of equivalent volume sphere. Both statements are not well mentioned in the manuscript, and we have clarified this in the revision. (Line 17 of Page 8 and Line 4 of Page 9)

3) How the coating has been associated with BC core is not clearly presented, are they partly coated or embedded? How did you treat the coating interaction with BC? One recent study (Liu et al., 2017, DOI: 10.1038/ngeo2901) could be referenced in page 6 line 10 or page 10 line 28 etc. to support your discussion.

Response: As shown in Figure 1(c), the BC core is totally embedded by the spherical coating. We simply introduce a coating sphere at the mass center of the BC aggregates, and the details of this model can be found in Liu et al (JQSRT, 2017). The interaction between the BC and coating is rigorously considered by the MSTM method, which is also one of the advantages of the model. To be more specific, with the inhomogeneous particle shape of Coated BC determined, the MSTM can consider the absorption and scattering properties of the particular particle accurately. (Line 23 of

Page 7 and Line 32 of Page 10) Liu et al. (DOI: 10.1038/ngeo2901) present an excellent work on the influence of coating and aging on BC optical and radiative properties, which is highly related to our work. The results of the paper may greatly benefit our studies, and we will study their results to get a better representation on Coated BC particles in further studies. Liu et al.'s work has been cited in the revised manuscript to support our discussion in the revision. (Line 16 of Page 6 and Line 16 of Page 7)

4) The empirical equation (equation 6) is almost all about refractive index uncertainty, and they are separately discussed for three different morphology cases. Though the refractive index has large variation from different literatures, but mostly we are using a fixed refractive index or fixed spectral dependence of refractive index, otherwise there will be no real value for anything. However, the authors have not really given how the BC morphology has actually influenced AAE, such as Df value, the amount of coatings associated. These are most interested to communities who care about how the BC ageing will influence its mixing state/morphology and how the AAE will be modified by these factors.

Response: BC morphology shows the most complicated influence on BC AAE, which can be quite different for particles with different sizes or refractive indices. As we can see from Figure 6, the AAE for BC at different sizes decreases to a totally different degree as BC becomes compact (from Fresh to Compact BC). Meanwhile, the influence of coating on the AAE would be much more complicated considering the realist particle geometries in the ambient atmosphere. Thus, we take the geometry as an independent factor for Equation 6, and give the empirical equation for each particle geometry. To qualitatively understand the effects of morphology, the AAEs of BC in the three different formats can be easily estimated by our empirical equations if its size and refractive indices are known, and, then, the influence of morphology can be derived. This means that we only consider the influence of BC morphology at certain particle size and refractive index, and the effects of geometry can be qualitatively given by the differences between two empirical equations representing particles with different geometries. We

add some discussions about the influence of BC morphology in the revision. (Line 24 of Page 1 and Line 26 of Page 13)

Others:

In page 6, the representation of coating thickness according to Schnaiter et al. (2005), could you point out which source of BC are they, and are they fresh BC, how long have they been aged?

Response: Thanks for the suggestion. The experiment given by Schnaiter et al. (2005) was carried out at a large aerosol chamber facility, and diesel soot particles were coated with secondary organic compounds produced by the in situ ozonolysis of - pinene. The particles are aged for 24 hours. We include additional informations in the revisions. (Line 10 of Page 7)

Page 6 line 30 to page 7 line 10, there are many parameter assumptions which have not been clearly explained: the 100 monomers are used, so are we actually only testing one BC core size? Have we tested the sensitivity to different monomer sizes (only 30nm is used here)? Liu et al., 2015 (DOI: 10.1002/2014GL062443) point out the AAE could be sensitive to the monomer size, also give the reference you choose 30nm.

Response: We considered the lognormal size distribution for BC particles, and clarified this point in the revision. Aggregates with 100 monomers are only an example to illustrate the particle geometries. As suggested by the reviewer, we have added the results with different monomer sizes in Figure 6. For Fresh BC with lacy structure, the monomer size doesn't change BC AAE significantly, whereas may decreases the AAE of Compact BC by approximately 0.1 as the monomer diameter increasing from 20 to 40 nm. Those discussions are also included in the revision. (Line 24 of Page 11)

For compact BC, the Df is used as 2.8 which is nearly sphere, any reference for this value? As above, it would be useful to test the sensitivity to Df.

Response: For compact BC, we considered the particles as compact as possible, and

thus a value of 2.8 is used in this study. To reveal the sensitivity of AAE to Df, and, thus, both a small (1.8) and a large (2.8) Df value are used in this study. Actually, there are not too many observations that give such a large Df, whereas some electronic microscopic images of BC show really compact structure. All other parameters are chosen based on observational data, and we had added the corresponding references. To better illustrates the sensitivity, we include one more curve in Figure 6 for results of aggregates with a Df of 2.3. For better understanding on the influence, the references for the Df values are included, and the sensitivity to Df is also discussed. (Line 3 of Page 12)

Page 7 line 8-16: the whole discussion here is rather confusing, you should point out what size previous instruments actually measured, the coated particle size or only BC core size, currently they are mixed up. You should point out SMPS measured mobility diameter is very sensitive to the particle shape (which is different from the volume equivalent diameter you present here), but references you referred Reddington et al., 2013; Wang et al., 2015 used the BC core size, measured by the single particle soot photometer.

Response: The discussion related to BC size is completely reorganized, and, with additional discussions, it should be easier to understand right now. (Line 9 of Page 8)

---

## Author Comment (AC3) · 26 Dec 2017

Overall: First of all, we would like to thank the two anonymous reviewers and Dr. Corbin for their thoughtful review and valuable comments to the manuscript. In the revision, we have accommodated all the suggested changes into consideration and revised the manuscript accordingly. All changes are highlighted in the revised manuscript in RED in the revision.

Inspecting aerosol AAE is important for multitude of applications, and the originally derived values are problematic, as the authors correctly point out.

The authors want to be able to break AAE (a derived particle-scale quantity) into, essentially, microphysical factors contributing to AAE, and this is indeed how I also see

the field moving forward. While derived quantities are useful in same situations, BC can be argued to be too varied, either due to coating or differences in aggregation, for a single number to reliably represent all cases.

I think the work done here is important and of high quality, and is definitely worth being published. I have a few main questions I'd like to see addressed as well as a few minor comments.

Response: Thanks for the comments on the manuscript. We definitely agree with the reviewer that there are significant uncertainties on BC AAE, and numerical models should make their contributions to improve our understanding. The following presents our answers as well as the revision for the manuscript.

Major notes:

Is the coating always observed to be a sphere encompassing the aggregate? For a less compact aggregate would the coating be a very large sphere or rather would the shape follow that of the aggregate? Is that something that could be studied with the model used here?

Response: Geometry is one of the most significant uncertainties on coated BC. In real atmosphere, the electronic microscopic images do show BC particles with quite different shapes. For less compact aggregates, some numerical models are developed to build coating with the shape following that of the aggregate, whereas none of those studies consider the AAE. We discuss some of those studies about different coated BC models in the revision. It is possible to consider almost any geometries numerically, and we tried out best to find the most representative one for general study. Meanwhile, we also stress that our work considers only a special case to account for the effects of BC aging. In the revision, we further emphasis the limitation of the Coated BC case, and further investigations should be carried out to consider the effects of other particle geometries. (Line 8 of Page 6)

[Figure]

Individual BC particles are not perfect spheres nor perfectly smooth in reality. Is it possible to investigate the effects while taking these non-idealized factors into account? Do the authors think these factors would have significant effects or can they safely be ignored?

Response: Yes, the non-idealized factors, such as overlapping or necking among monomers, nonsphericity, and poly-disperse monomer, do exist in reality, and even the fractal aggregate model may show differences from the realistic BC cluster. Those effects can be considered by numerical models other than the MSTD, and there are some studies that investigated such effects, such as Skorupski and Mroczka (2014), Yon et al. (2015), Wu et al. (2016a, 2016b), and Liu et al. (2016). These studies generally indicate that the minor factors do influence the optical properties to some degree, whereas those influences on the absorption are minor compared to the particle overall geometry or size. Overall, the size and overall geometries show the most significant influences on BC optical properties, and, thus, this study captures those main factors. We include similar discussions in Section 2.1 of the revision. There will be definitely following-up studies to show the influences of those factors on BC AAE, considering that the effects of those factors on BC AAE have not been considered yet. (Line 20 Page 5)

Is there a way to parametrize the results by using e.g. equivalent-sized (coated) Mie spheres with some kind of an effective medium approximation? This would greatly help facilitate the use of these types of particles in many applications (moreso than just a better estimate of AAE).

Response: This is a great suggestion for the approximation. As we have seen from other studies, neither the equivalent-sized Mie sphere nor equivalent medium approximation can give accurate approximation on optical properties of BC aggregates (Li et al., 2009; Liu et al., 2013). We included those previous studies in the revision. Furthermore, in Figure 6 of the revision, we include results given by Mie for the equivalent volume sphere approximation, and discusses the errors that may be introduced. As we

can see from the figure, the Mie results are significantly different from others. Considering the poor performance given by the Mie results, we will not include the results for core-shell Mie or for equivalent medium approximation, because the inhomogeneous structure will bring another factor for the numerical errors. The changes are made at Figure 6 and the corresponding discussions. (Figure 6 and Line 19 of Page 11)

Minor notes:

Page 9, line 16: "straightforward", not "straight forward"

Response: Corrected. (Line 24 of Page 10)

Page 10, line 13: Reference is italicized, unlike most other references. Should be consistent

Response: Thanks, and we have modified the format. (Reference Section)

Fig. 2: X-axis says "Evquivalent", should be "equivalent"

Response: Corrected. (Figure 2)

Fig. 9: Legend says "Comapct" instead of "Compact" Response: Corrected. (Figure 9)

---

## Author Comment (AC4) · 26 Dec 2017

Overall: First of all, we would like to thank Dr. Corbin and the two anonymous reviewers for their thoughtful review and valuable comments to the manuscript. In the revision, we have accommodated all the suggested changes into consideration and revised the manuscript accordingly. All changes are highlighted in the revised manuscript in RED in the revision.

The manuscript "The absorption Angstroem exponent of black carbon from numerical aspects" by Liu, Chung and Yin (ACPD 2017, doi:10.5194/acp-2017-836) is on a very interesting topic. The authors present numerical simulations of BC properties for 3 test cases of morphology using computer-modelled soot aggregates.

[Figure]

Response: Thanks Dr. Corbin for the constructive comments on the manuscript, and we revised the manuscript following all the suggestions

The value of these results is clear; the effects of variations in BC microphysical properties on the AAE are clearly seen. However, the limitations of the results could be discussed in much more depth. I have several comments related to this:

1. The words "diameter" and "size" are used in multiple ways throughout the study. Especially when discussing the results of Schnaiter et al., with respect to coating thicknesses of BC, the word diameter is poorly defined as it would refer only to the apparent mobility of the particles (increased by coatings but decreased by restructuring).

Response: During the revision, we improved the clarity of the terms "diameter" and "size" by adding exact definition for diameter, and detailing and improving the discussion about the coating thickness. The definition for BC size can be quite different considering the different principles for size measurements. In this study, we try to keep the discussion simple, and both diameter and size refer to the diameter of equivalent volume sphere for consistency. We clarified this definition in the revision, and, in the section for the coating thickness, we illustrate the definition of three quantities in the figure and specify each of them clearly in the discussion, which will help readers to better understand the discussion. In this revised version, there will be little chance for misunderstanding the definition of diameter we used. (Line 16 of Page 8, Line 30 of Page 6, and Figure 2)

2. The use of a change in mobility diameter (Schaiter et al.) to infer size-dependent coating thicknesses is invalid, as noted above. As a reviewer noted, Liu et al. (Nat. Geosci 2017) should be referenced here. In any case, I am not sure of the value of this size-dependent coating thickness. Since all of the results were plotted against diameter, there is no possibility that the reader misunderstands that coating thickness depends on particle size and history. A constant volume fraction of coatings would allow a fair evaluation of the size dependence to be made. Currently the x axis of

the figures (labelled GMD) reflects changes in both size and coating thickness, and is therefore difficult to interpret.

Response: First, our paper discusses the relationship between the coating thickness and the BC core size. To be clear, the paper does not discuss how this relationship changes over time as the coating progresses. What is discussed in the paper is a snapshot relationship after 24 hours of coating. Such a snapshot relationship (whether after 24 hours or after 48 hours) is what we need for our study.

Many papers do provide the size distribution of coated particles, but Schnaiter et al. (2005) provide the size distribution of BC cores as well. In reality, Schnaiter et al. (2005) provide the size distribution of coated particles and that of fresh BC. Since a fresh BC particle becomes a BC core after coating, both the fresh BC and the BC core have the same diameter of the corresponding equivalent volume. Schnaiter et al. (2005) actually employed an SMPS (Scanning Mobility Particle Sizer) to measure the particle sizes, and SMPS gives mobility diameter. Mobility diameter is very close to the diameter of the corresponding equivalent volume for spherical particles (and thickly coated BC are nearly spherical) but mobility diameter may deviate from the diameter of the corresponding equivalent volume for fresh BC. However, without information for particle shape and density, it is difficultly get the relationship between the two diameters, and we made this assumption in the study. (Line 30 of Page 6)

A constant volume fraction of coating assumes that small BC cores have thin coating and large BC cores have thick coating, but there is little backing evidence for this assumption. Originally, we thought of a constant volume fraction of coatings too, but later realized that this assumption may be wrong. The relationship between coating thickness and core size is developed in our paper with two ideas: (1) We try to derive and use a more realistic relationship between coating thickness and core size that has an observational basis. Although the derived relationship in our paper is based only one experimental study (i.e., Schnaiter et al. 2005) and used mobility diameter data, it is the first derived relationship in the community that has any observational support. (2).

As we have emphasized in the manuscript, we understand the results for Coated BC as a special case, and much more work should be done to give a more general conclusion considering the complex properties of Coated BC. Thus, we prefer to keep using the coating thickness we obtained, but better clarify the discussion in the revision.

Liu et al. (2017) presented a state-of-art study on the BC absorption enhancement due to particle mixing. Their work can definitely improve our understanding on the relationship between BC core size and coating, and we would try to obtained such relationship for sensitive studies in the future. (Line 16 of Page 6 and Line 16 of Page 7)

3. The number of primary particles in an aggregate was mentioned as hundreds or even thousands, but what is the most likely number?

Response: We consider BC aggregates with different monomer number, and only the bulk properties averaged over given size distributions are discussed. As we have demonstrated in the manuscript, we consider aggregates with monomer number from 1 to 2000. To know the most likely number, we can give an example here. With monomer diameter defined as 30 nm, an aggregate with approximately 300 monomers corresponds to an equivalent volume diameter of 0.2 m. The most likely number could vary from case to case, and, thus, we consider a relatively wide range of size distribution for sensitivity study. We have also give more detailed relationship between monomer number and equivalent-volume diameter in the revision. (Line 32 of Page 8)

4. (page 6) The authors might cite the previous studies which have modelled coatings on complex BC morphologies, e.g. Liu et al. 2016 (doi:10.1016/j.jqsrt.2015.08.005) and references therein. As the authors noted, however, no previous study discussed the AAE.

Response: Yes, we added some discussion about this point in the revision at Line 29-34, Page 6, and Liu et al. (2016) as well as Dong et al. (2015) and Liu et al. (2012) is cited now. (Line 8 of Page 6)

5. To what degree might the results change with different choices of monomer diameter and monomer number?

Response: In the revision, we include the results for particles with different monomer diameter. As shown in the updated Figure 6, the AAE increases as the monomer diameter deceases. The monomer diameter is not a very big problem for Fresh BC, but becomes important for Compact BC. Meanwhile, the variation over monomer number, i.e. particle overall size, can be understood by that over the GMD in this study. With the increases of monomer number, the GMD increases, and the AAE decreases. (Figure 6 and Line 24 of Page 11)

6. On page 11, lines 10-14 do not follow from the previous discussion. The authors are saying that the BC RI may change between sources, so that the AAE may not be fixed. But it has been assumed that the imaginary and real RIs are free to vary independently, which is not a justified assumption (Bond et al. 2006; Moteki et al. JAS 2010, doi:10.1016/j.jaerosci.2010.02.013). This assumption would have led to overestimated variability in AAE. Also, this assumption was necessary to define A and B as independent in equations 4 and 5.

Response: The discussion is unappropriated here in the discussion, and we omitted them in this paragraph. Although the real and imaginary parts cannot vary independently, and this study defines A and B independently to better understand the effects of each parameter on BC AAE. For realistic applications, A and B can be obtained based on the given wavelength-dependent refractive indices. (Line 20 of Page 10)

Also I have a couple of minor comments which were not noted by the reviewers:

• Palas soot (page 3) should be referred to as spark-generated carbon nanoparticles.

Response: We modified the text following the suggestion. (Line 12 of Page 3)

• At page 4 line 20, Moosmüller et al. (2011, doi:10.5194/acp-11-1217-2011) should

be cited.

Response: This is a really great paper to support our work, and we cited it in the revision. (Line 4 of Page 4)

• In Section 2.2, please rewrite the discussion. It currently sounds like you are saying that an SMPS measures the diameter of volume equivalent spheres.

Response: SMPS measures the mobility diameter of the particle, and we clarified this point in the revision. (Line 9 of Page 8)

• On page 10, I don't understand why a linear regression (I assume of log- transformed data?) should give the best representation of the AAE if the relationship is not exponential. A linear regression of log-transformed data is not reliable even when the data follow a power-law relationship (Clauset et al., 2009, doi:10.1137/070710111)

Response: The linear regression is applied for the log-transformed data, i.e., for the linear relationship shown in Figure 5. Considering the highly agreement between the simulated data and the fitting, the method to get the fitted data doesn't influence the results too much. We have clarified that the data is obtained by linear regression of the log-transformed data. (Line 7 of Page 11)

Finally, the authors have proposed that their numerical results are a suitable absolute reference for BC properties. This is clearly not the case, as a closure between experiment and theory is still lacking (Bond et al., 2006; Radney et al., 2014). Radney et al. (2014, doi:10.1021/es4041804) is particularly relevant to this study because they have used the same T-matrix approach in combination with direct measurements of uncoated BC, yet were unable to satisfactorily model particle extinction. This shows that T-matrix models of BC aggregates should not be used to make strong conclusions on BC properties. Rather than interpret these numerical results as a comprehensive guide and absolute reference, the authors could interpret these results as a clear demonstration of the importance of various parameters on the optical properties of BC. As noted

above, I found these results to be quite interesting, because they demonstrate quantitative the theoretical impacts of changes in various properties on BC in a way that has not been done before.

Response: Thanks for the suggestion, and we agree with Dr. Corbin on the limitation of the numerical models and the suggested interpretation. Although the results can hardly be understood as an absolute reference for BC AAE, we think it is fine enough to illustrate the response of BC AAE to BC size, refractive index, and geometry, at least qualitatively. Thus, we have incorporated these into the revised version, and discussed the corresponding studies. (Line 21 of Page 15)

References

All references were either made in the submitted manuscript or were cited using doi's. These doi's provide direct links to the articles via http://dx.doi.org

Response: Thanks again for showing us those important references, and most of them are helpful for our work and discussed in the revision.

---

## Author Response (AR2)

**Responses to Reviewers (ACP Manuscript # ACP-2017-836)**

First of all, we would like to thank Dr. Corbin for his re-review on the manuscript. In the revision, we followed both major suggestions and significantly improved the manuscript, especially for the derivative related to coating amount. All changes are highlighted in RED. Meanwhile, in this point-to-point response, the reviewers' comments are copied as texts in BLACK, and our responses are followed in BLUE.

The Editor has requested my feedback on whether the revised manuscript sufficiently addressed my comments. Unfortunately, I cannot answer this question in the affirmative. I would define "sufficiently addressed" as either (i) disagreeing with a comment and providing an evidence-based argument for such disagreement, or (ii) agreeing with a comment and providing a modified manuscript which remains self-consistent. In the two most substantial cases, it appears that the authors have agreed with my comments, but not modified the manuscript in response. Therefore, I believe that major revisions to the revised manuscript would be required for it to send a self-consistent and unambiguous message.

Of my six comments, three addressed a lack of literature context (#3, 4, 5), one (#1) addressed the inaccuracy with which the manuscript uses technical language, and two (#2, #6) addressed the degree to which the manuscript contributes to scientific progress. The latter two (#2 and #6) are the ones I consider most substantial, although I do not see that #1 was addressed sufficiently either.

Regarding #2, my comment related to the fact that the authors have based their calculations on an invalid reinterpretation of literature results (the assumption that a change in mobility diameter can be used to infer coating thicknesses). In my original comment I provided a solution for the authors to avoid this problem without much more work. The modified manuscript has not changed this assumption, although the authors appear to agree with my comment (there is a new statement that "the conclusions must be viewed with some caution" on page 7). This makes the manuscript appear non-self-consistent.

**Response**: We agree with the reviewer that the assumption of using mobility diameter for fresh BC is problematic. To avoid such error, we made a more rigorous calculation to translate the mobility diameter into the equivalent volume diameter for both BC core and

Coated BC, and, then, the observation and our numerical model can be connected more accurately. To be more specific, this is done in the following ways:

(1). For Coated BC, we assume the mobility diameter to be a reasonable representation of equivalent volume diameter, because they are compact and nearly spherical.

(2). For Fresh BC, instead of directly using the mobility diameter from the SMPS, we first transfer the mobility diameter into equivalent volume diameter (Fig. 2b) following the relationship between mobility diameter and aggregate parameters given by Naumann (2003) and the diesel soot geometric parameters given by Schnaiter et al. (2003; 2005). In this way, we significantly improve the accuracy for the description of the Fresh BC (i.e. BC core sizes) size distribution.

With the simplifying assumptions that are much more reasonable, we can get the volume fraction of coating as a function of equivalent volume diameter of BC core, which is what we need for our study and not conflict with our aggregate models. As shown in Figure 2d, the volume fraction of coating decreases as BC core size increases. This coating amount variation we obtained agrees with the trend obtained by Fierce et al. (2016). This also indicates that our coating may be not quantitatively perfect, but is qualitatively right, at least in the right direction.

To further address this concern on the coating amount distribution, we perform additional simulations for Coated BC with fixed coating thickness and fixed coating volume fraction for comparison. The following figure illustrates absorption enhancement due to coating with different coating assumptions (i.e., the one we derived, fixed coating fraction, and fixed coating thickness (based on equivalent volume sphere)). For all three cases, the total coating amount for the ensemble of different-sized BC particles is the same, whereas the coating distribution is different for BC cores with different core sizes. We can see that three cases lead to similar absorption enhancement, so we conclude that our results based on the derived coating amount should be similar with those based on fixed coating fraction. However, our results are supported by the observations.

To conclude, by mainly changing the mobility diameter into equivalent volume diameter for Fresh BC, we have significantly improved the discussion about the coating derivation (see entire Page 7 of the revision), and reconstructed the Figure 2 with more details. The coating amount given should be much improved and reasonable.

Furthermore, we would really like to continue work on this task based on Liu et al. (Natrue Geoscineces, 2017) to derive more reasonable and generally coating thickness dependence. Meanwhile, we have demonstrated in the text that the results should be carefully understood as those from a special case.

[Figure]

Figure 1. Absorption enhancement due to non-absorption coating with different coating assumptions, and the totalThe lognormal size distribution with a GMD of 0.12 micron and a GSD of 1.5 is assumed for the BC core.

Regarding #6, my comment related to the treatment of the real and imaginary parts of the complex RI as independent being inconsistent with literature. The authors appear to agree (according to their adding a similar statement on page 10), but have not altered the analysis as a result. Again, this makes the manuscript appear not to be self-consistent. Reading between the lines, I believe that the authors intended for their calculations to demonstrate the impact of uncertainty in the RI values. This I would understand, but according to the current revision I cannot be sure that this is what the authors intended. If it is, then it should be clearly stated and the entire manuscript adjusted accordingly.

**Response**: Yes, we agree with the reviewer that the wavelength variation on the real and imaginary parts of refractive indices (i.e. parameters A and B in the manuscript) should be dependent, and we add extra discussions to stress this point to keep the manuscript consistent. In the revision, we kept Figure 8 to illustrate the uncertainties related RI wavelength dependences, whereas the quantitative discussion on A and B based on Equation (6) is removed. Because independent A and B may overestimate the variability of BC AAE, Equation (6) is simplified to consider only wavelength-independent refractive index. The modifications are made at Lines 5-9 of Page 11, Lines 13-14 of Page 14, and Line 33 of Page 14 – Line 2 of Page 15, and we also remove some statements that may confuse the readers as suggested by Dr. Corbin.

Because the reviewer thought that some other comments from the original review are not addressed sufficiently, the following lists the original comments #1 and #4 as well as the response.

1. The words "diameter" and "size" are used in multiple ways throughout the study. Especially when discussing the results of Schnaiter et al., with respect to coating thicknesses of BC, the word diameter is poorly defined as it would refer only to the apparent mobility of the particles (increased by coatings but decreased by restructuring).

**Response**: The section related to the coating amount is corrected and totally rewritten, and we use the "diameter" and "size" in the revision really carefully. For coating, we consider only coating amount in this revision, and there will be much less confusing statements.

4. (page 6) The authors might cite the previous studies which have modelled coatings on complex BC morphologies, e.g. Liu et al. 2016 (doi:10.1016/j.jqsrt.2015.08.005) and references therein. As the authors noted, however, no previous study discussed the AAE.

**Response**: The reference is added, and we further specify the uniqueness of our study in the revision.

Last but not the least, we thank Dr. Corbin again for the constructive comments, and both suggestions have significantly improved our paper.

---

## Author Response (AR3)

Dear Drs. Laaksonen and Corbin,

Thanks again for Dr. Corbin's careful review on the paper. There is only a minor comment on the manuscript, and we have corrected according the suggestion.

Minor comment: on page 14 lien 2, please replace "size" with "diameter" or "radius" depending on which meaning was intended.
Response: Corrected.